# DSRF: A Dynamic and Scalable Reasoning Framework for Solving RPMs

**Chengtai Li**[1,2]  **Yuting He**[1,3]  **Jianfeng Ren**[1,4*]  **Ruibin Bai**[1,4]  **Yitian Zhao**[2]  **Xudong Jiang**[5]

[1]The Digital Port Technologies Lab, School of Computer Science,
University of Nottingham Ningbo China
[2]Cixi Institute of Biomedical Engineering,
Ningbo Institute of Materials Technology and Engineering, Chinese Academy of Sciences
[3]Shenzhen Institute of Advanced Technology, Chinese Academy of Sciences
[4]Beacons of Excellence Research and Innovation Institute,
University of Nottingham Ningbo China
[5]School of Electrical and Electronic Engineering, Nanyang Technological University

{scxcl2, scxyh2, jianfeng.ren, ruibin.bai}@nottingham.edu.cn
yitian.zhao@nimte.ac.cn, exdjiang@ntu.edu.sg

## Abstract

Abstract Visual Reasoning (AVR) entails discerning latent patterns in visual data and inferring underlying rules. Existing solutions often lack scalability and adaptability, as deep architectures tend to overfit training data, and static neural networks fail to dynamically capture diverse rules. To tackle the challenges, we propose a Dynamic and Scalable Reasoning Framework (DSRF) that greatly enhances the reasoning ability by widening the network instead of deepening it, and dynamically adjusting the reasoning network to better fit novel samples instead of a static network. Specifically, we design a Multi-View Reasoning Pyramid (MVRP) to capture complex rules through layered reasoning to focus features at each view on distinct combinations of attributes, widening the reasoning network to cover more attribute combinations analogous to complex reasoning rules. Additionally, we propose a Dynamic Domain-Contrast Prediction (DDCP) block to handle varying task-specific relationships dynamically by introducing a Gram matrix to model feature distributions, and a gate matrix to capture subtle domain differences between context and target features. Extensive experiments on six AVR tasks demonstrate DSRF's superior performance, achieving state-of-the-art results under various settings. Code is available here: https://github.com/UNNCRoxLi/DSRF.

## 1 Introduction

Abstract Visual Reasoning (AVR) aims to identify latent patterns from visual clues and then derive underlying rules [1]. It has been widely used for human IQ tests [2]. Recently, AVR has become a common benchmark for evaluating the reasoning ability of LLMs [3–5]. Among various AVR tasks, Raven's Progressive Matrices (RPMs) are most popular [6–10], where the question set is typically a 2×3 or 3×3 image matrix, with each row/column following a common rule and the bottom-right slot to be filled in. RPMs reduce language and cultural barriers by creating rules using symbols and geometric shapes, requiring pure reasoning ability without relying on prior knowledge.

---

*Corresponding Author

39th Conference on Neural Information Processing Systems (NeurIPS 2025).

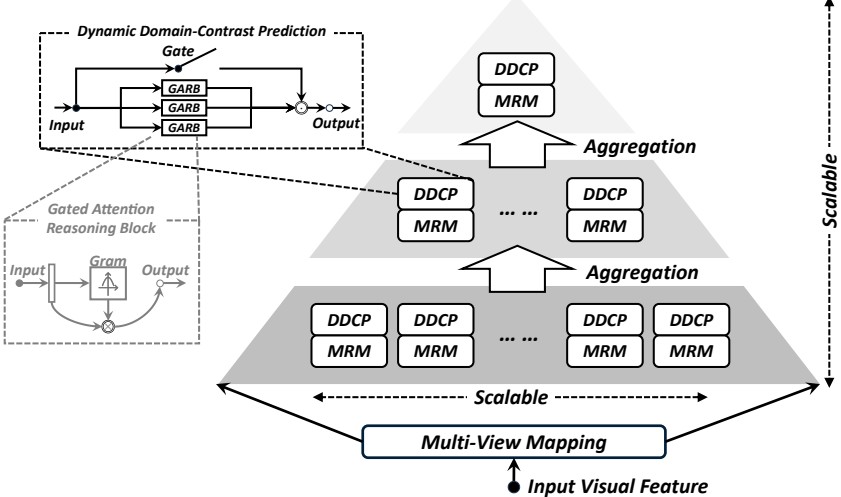

Figure 1: The DSRF is **scalable**, as it maps high-level abstract visual features into multiple views and extracts complex rule from these views through a layered pyramid, and it is **dynamic**, as each DDCP is equipped with various dynamic mechanisms to capture task-specific rules within each view.

Existing RPM solvers [9, 11–17] often establish relations between images through a perception module to encode visual features and a reasoning module to capture underlying rules. Reasoning modules are often built from shallow networks, which are much smaller than perception modules, greatly limiting their reasoning ability. As the visual features extracted from AVR tasks are rather abstract, and the reasoning rules are often concise and intuitive, stacking deep reasoning modules may increase the risk of overfitting [18]. Recently, a deeper reasoning network is introduced by incorporating reasoning blocks at multiple perceptive fields and aggregating the reasoning results from multiple layers [19]. However, shallow visual features lack the information needed for effective reasoning. The lack of scalability of reasoning networks compared to the well-established scalability in visual perception networks greatly limits the ability of existing RPM solvers.

To tackle this challenge, we propose a **M**ulti-**V**iew **R**easoning **P**yramid (**MVRP**), a scalable layered reasoning pyramid. As shown in Fig. 1, the multi-view mapping enables the model to easily scale up to cover more attribute combinations analog to reasoning rules. In addition, rules in RPMs are often human-understandable, concise and intuitive. Deep reasoner in existing models [15, 19] may potentially overfit the training data. Furthermore, images often contain a huge number of feasible rules across diverse attributes, *e.g.*, Shape, Size, Position, Color, and Number [6, 20]. To effectively capture these rules, instead of deepening the network, the proposed MVRP focuses on widening the network to model more complex rules.

Another challenge is that previous methods [9, 15, 17, 19] often employ static reasoning networks built from convolutional neural networks (CNNs) or multi-layer perceptrons (MLPs), in which static filters and network weights are derived from the training data and faithfully applied to novel test samples, expecting a minimum domain shift. These static reasoning modules greatly limit their adaptability to task-specific demands in novel application scenarios. Thus, it is critical to design a dynamic task-adaptive mechanism to flexibly handle diverse rule sets in novel test scenarios.

To dynamically model underlying relations and strengthen reasoning ability over tremendous rules, we propose a **D**ynamic **D**omain-**C**ontrast **P**rediction (**DDCP**) block, with three dynamic aspects in network design. 1) The gating mechanism in DDCP, helping the network adapt to various input features and automatically highlight relevant features for rule construction. 2) Dynamic modeling of domain differences by a novel **G**ated **A**ttention **R**easoning **B**lock (**GARB**). A Gram matrix derived from self-correlation is designed to model the distribution of context features and another for target features. The GARB dynamically contrasts the domain difference in three aspects, and it is in turn applied to highlight the context features for better constructing the rules. As shown in [21], the GARB allows our model to flexibly adjust its parameters following the input, thereby improving its ability to handle varying conditions. 3) Dynamic gating in GARB for highlighting domain difference. The

domain difference is complex and diverse, in which irrelevant features may mislead the reasoning process. The dynamic gate highlights the domain difference related to context features and target features, adaptively focusing on relevant features for establishing the underlying rules.

Our contributions can be summarized as follows: 1) The proposed **D**ynamic and **S**calable **R**easoning **F**ramework (**DSRF**) solves RPMs in a scalable and dynamic manner. 2) The proposed MVRP well captures complex rules through a layered reasoning pyramid, which could scale up the network easily by mapping high-dimensional features into different views. 3) The proposed DDCP greatly enhances the generalization capability of the reasoning networks through dynamic domain modeling and dynamic gating designed in GARB. 4) Extensive results on six AVR datasets show that DSRF significantly outperforms state-of-the-art models and generalizes well on novel tasks.

## 2  Related Work

Abstract Visual Reasoning has recently attracted growing research interests. In particular, a lot of efforts have been made to enhance the reasoning capabilities of large language models [3, 4]. AVR tasks can be broadly categorized as follows [1]. 1) RPM tasks challenging the subject to identify the right answer according to the underlying rules in a question panel, including RAVEN [6], I-RAVEN [8], RAVEN-FAIR [9], PGM [7], Unicode [10], and many others [8, 9]. 2) Same-different tasks such as SVRT [22] to discern similarities and differences according to underlying rules. Recently, it is extended to the CVR task [20] and $MC^2R$ task [18], where models must identify one or more outliers from a group of images. 3) Other research directions in AVR, *e.g.*, Bongard Problems [23] consisting of abstract shapes and rules for assessing reasoning methods in scenarios with limited samples; Arithmetic Visual Reasoning [24] combining numerical understanding with visual elements; Relations Game [25] centered on detecting explicit spatial-relational rules among objects in images.

AVR models often consist of a perception module and a reasoning module. As most AVR tasks are built from regular shapes, relatively shallow networks are often employed in AVR models, *e.g.*, RelBase [12], MRNet [9], PredRNet [15], AlgebraicMR [13] and SCAR [17]. Recently, in view of excellence of transformers in capturing the long-range global interaction, HCV-ARR [14], DRNet [16] and $HP^2AI$ [19] incorporate both transformer branch and CNN branch to enhance visual features.

Traditionally, AVR models are often designed to capture the row- and/or column-wise similarities using addition, subtraction, and dot product operations, *e.g.*, MRNet [9], DRNet [16] and HCV-ARR [14], or establish the weights for images to model the underlying relations [12, 13, 17]. Recent works focus on embedding the abstract rules into the reasoning network through a "predict-and-verify" paradigm [15, 19, 18]. Despite the rapid developments, existing AVR models often lack the scalability to boost their reasoning capability, *e.g.*, both MRNet [9] and HCV-ARR [14] focus on capturing row and column similarities, leading to a shallow reasoning network that greatly restricts the scalability. Recently, PredRNet [15] and $HP^2AI$ [19] scale the network by stacking the reasoning blocks, their models have the problem of overfitting after stacking just four reasoning blocks. The lack of scalability greatly hinders the development of more general and more powerful reasoning models. Furthermore, existing models such as PredRNet [15], $R^3PCL$ [18], DBCR [26], DARR [27] and $HP^2AI$ [19] often fit a static reasoning network to underlying rules, limiting the reasoning ability and generalizing poor on novel data. In this work, we propose DSRF to tackle the challenge of dynamically and comprehensively reasoning over complex rules in an easily scalable manner.

## 3  Proposed DSRF

### 3.1  Overview of Proposed DSRF

Formally, the RPM problem is defined as $\langle Q; A \rangle$, where $Q$ is the question matrix, typically consisting of 3×3 or 2×3 images of size $H \times W$ excluding the last slot, and $A$ is the set of candidate answers. We take the most popular AVR task, the RAVEN problem [6] of $3 \times 3$ images as an example, where $Q = \{Q_1, Q_2, \ldots, Q_8\}$ and $A = \{A_1, A_2, \ldots, A_8\}$. The target is to select one image from $A$ to fill in the missing slot of the question panel, $\hat{y} = \mathcal{R}_{\Theta_r}(\mathcal{P}_{\Theta_p}(\langle Q; A \rangle; \Theta_p); \Theta_r)$, where $\hat{y}$ is the predicted answer, $\mathcal{R}_{\Theta_r}(\cdot)$ and $\mathcal{P}_{\Theta_p}(\cdot)$ are the reasoning module and perception module, respectively.

The overview of DSRF is shown in Fig. 2. We utilize four ResNet blocks [15] as the perception module. Other backbones can also be applied as shown in Sec. 4.3. For scalable reasoning, the

proposed MVRP incorporates a layered pyramid to encapsulate the composition structure of complex rules, and a multi-view mapping to provide a convenient mechanism to map features into different views, easily widening the network to cover tremendous complex rules. To handle the domain

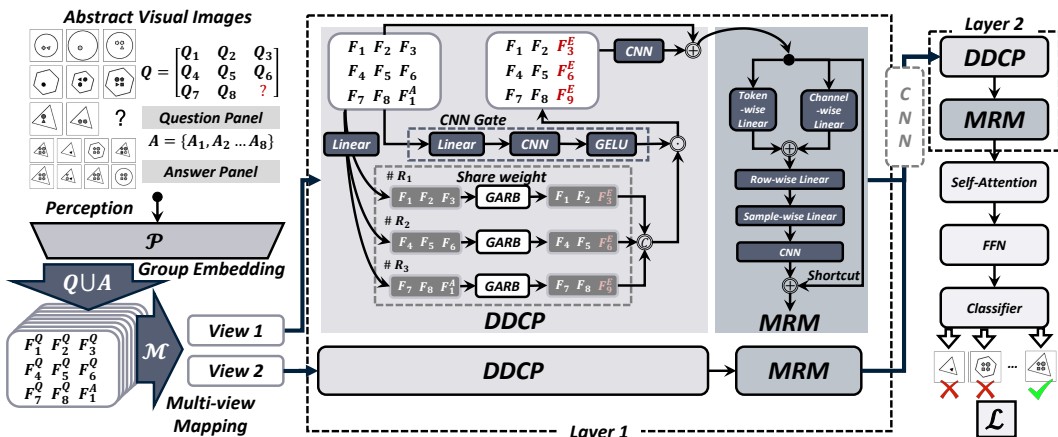

Figure 2: Overview of proposed DSRF-Small. Visual features are mapped into multiple views, and a DDCP block highlights the domain differences and dynamically encapsulates the underlying rules through GARBs in each view. The MRM explores rule combinations across different levels of granularity. All the views are fused and passed to the next pyramid level, progressively refining rules.

drift between context and target images, a DDCP block is designed in each view, where a GARB encapsulates the abstract rules under the "predict-and-verify" paradigm [18], and highlight the domain difference through dynamic gating and dynamic domain contrast. Additionally, a convolution gate [28] is designed to dynamically highlight the most relevant rules in a view. To better comprehend complex rules, a multi-granularity rule mixer (MRM) is designed to explore rule combinations through channel-wise, token-wise, sample-wise and row-wise feature mixing.

## 3.2 Multi-View Reasoning Pyramid

**Multi-View Mapping:** The MVRP is structured as a layered reasoning pyramid, where each layer employs a multi-head mechanism to map the features into different views, each representing a different group of attributes. As a result, different views focus on distinct combinations of attributes, and complex rules can be constructed layer-by-layer through the reasoning pyramid. More importantly, by mapping into more views, the proposed method can easily enlarge the network to capture rich rules. Specifically, consider layer $l$ in the pyramid with $H$ views, the mapping $\mathcal{M}$ is defined as,

$$\{\boldsymbol{V}_1^l, \boldsymbol{V}_2^l, ..\boldsymbol{V}_H^l\} = \mathcal{M}(\boldsymbol{X}^l), \boldsymbol{X}^l \in \mathbb{R}^{T \times C^l}, \boldsymbol{V}_h^l \in \mathbb{R}^{T \times C^h}, \tag{1}$$

where $\boldsymbol{X}^l$ represents the input features of layer $l$ and $\boldsymbol{V}_h^l$ represents the $h$-th view of layer $l$, $T$ is the token size, $C^l$ and $C^h$ denote the number of channels at layer $l$ and the $h$-th view, respectively.

**Reasoning Pyramid:** The layered pyramid built from multi-view features aims to encapsulate the complex reasoning rules. Formally, denote $\mathcal{F}_{\text{DDCP}}^{l,h}(\cdot)$ as the function of DDCP for the $h$-th view at layer $l$, which dynamically analyzes the domain difference between context and target features in each view. Denote $\mathcal{F}_{\text{MRM}}^{l,h}(\cdot)$ as the function of MRM, which investigates rule combinations via multi-granular feature mixing. The output features for the $h$-th view at the $l$-th layer are obtained as,

$$\hat{\boldsymbol{V}}_h^l = \mathcal{F}_{\text{MRM}}^{l,h}(\mathcal{F}_{\text{DDCP}}^{l,h}(\boldsymbol{V}_h^l)). \tag{2}$$

We aggregate the features from all views for layer $l$ as $\boldsymbol{P}^l = \sum_{h=1}^{H} \boldsymbol{W}_h^l \hat{\boldsymbol{V}}_h^l$, where $\boldsymbol{W}_h^l$ represents the weight matrix for the $h$-th view at layer $l$. By aggregating the multi-view features, the model captures a rich set of rule combinations from different views. Then, as the features pass through consecutive layers, the model gradually improves its understanding of complex rules.

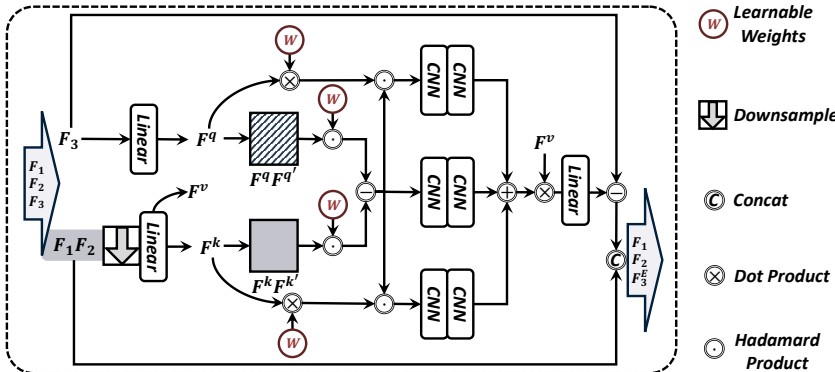

Figure 3: Block diagram of GARB. Techniques including Gram matrix for modeling feature distribution, learnable weights, gating mechanism and attention mechanism enhance dynamic reasoning.

**Discussion of Scalability:** Deep networks, while powerful, are prone to overfitting [29, 30]. Typically as abstract rules are concise and intuitively defined for reasoning tasks, deep reasoning networks may easily overfit [15, 19]. Instead of common practice of deepening the network in visual recognition, the MVRP expands the network by scaling up the number of views to encompass a broad set of attribute combinations. As shown later in the experiments, the model generalizes better on Out-Of-Distribution (OOD) settings and reasons better over the complex rules in many datasets.

Specifically, we design three pyramids with small, medium and large configurations, with a pyramid structure of "2-1", "4-2-1", and "8-4-1", where each number represents the number of views in each layer. This progressive increase of views enables scalability by covering more combinations of attributes. As shown in experiments, all three models, **DSRF-S**, **DSRF-M**, and **DSRF-L**, achieve excellent reasoning accuracy across various settings on different datasets. The large model DSRF-L performs best without overfitting, but at a higher computational cost.

### 3.3 Dynamic Domain-Contrast Prediction Block

The proposed DDCP block is built on the "predict-and-verify" paradigm [18]. Given features $\{\boldsymbol{F}_i \in \mathbb{R}^{T \times C^h}\}_{i=1}^3$ for the three images, we treat the first two $\boldsymbol{F}_1, \boldsymbol{F}_2$ as the context and $\boldsymbol{F}_3$ as the target, the goal is to derive a prediction function $\mathcal{R}$ that minimizes the prediction error, $\boldsymbol{F}_3^E = \boldsymbol{F}_3 - \mathcal{R}(\boldsymbol{F}_1, \boldsymbol{F}_2)$,

$$\arg\min_{\mathcal{R}} ||\boldsymbol{F}_3 - \mathcal{R}(\boldsymbol{F}_1, \boldsymbol{F}_2)||^2. \tag{3}$$

By doing so, $\mathcal{R}$ could progressively capture the underlying rules. Fig. 2 shows that Gated Attention Reasoning Block (GARB) serves as the core reasoning module, which leverages self-correlation to model the distributions of context and target features for dynamic reasoning. Note that for RAVEN problems [6], the same reasoning network $\mathcal{R}(\cdot)$ is applied across three rows to ensure the rule consistency. Inspired by [28], to further highlight the critical rule information, we employ a convolutional gate as shown in Fig. 2 to selectively emphasize relevant features while suppressing noisy or irrelevant rules. As a result, DDCP enhances the model's adaptability and generalization by allowing dynamic rule hypothesis and validation, effectively handling complex and diverse rule sets.

### 3.4 Gated Attention Reasoning Block

The proposed GARB is the core reasoning module to dynamically reason over a large number of rules. As shown in Fig. 3, we resort to the attention mechanism [31–33] and gate mechanism [31, 28] to enhance the dynamic nature of the reasoning module. For illustration, we use the first row as an example. We first linearly project the target features and context features as in [28],

$$\boldsymbol{F}^q = \mathcal{F}_l^q(\boldsymbol{F}_3), \quad \boldsymbol{F}^d = \mathcal{F}_{\text{DS}}(\{\boldsymbol{F}_1, \boldsymbol{F}_2\}), \quad \boldsymbol{F}^k = \mathcal{F}_l^k(\boldsymbol{F}^d), \quad \boldsymbol{F}^v = \mathcal{F}_l^v(\boldsymbol{F}^d), \tag{4}$$

where $\boldsymbol{F}^q$, $\boldsymbol{F}^k$, and $\boldsymbol{F}^v$ are query, key, and value matrices, respectively, $\mathcal{F}_l^q$, $\mathcal{F}_l^k$ and $\mathcal{F}_l^v$ are three corresponding linear projections, and $\mathcal{F}_{\text{DS}}$ is a Down-Sample function.

Following [34], we utilize the Gram matrix derived from the self-correlation to model the distribution of context features and target features, and derive the domain-difference features $\boldsymbol{A}^d \in \mathbb{R}^{T \times T}$ as,

$$\boldsymbol{A}^d = \frac{1}{\sqrt{C^h}} \left( \sigma(\boldsymbol{W}^q) \odot (\boldsymbol{F}^q \boldsymbol{F}^{qT}) - \sigma(\boldsymbol{W}^k) \odot (\boldsymbol{F}^k \boldsymbol{F}^{kT}) \right), \tag{5}$$

where $\odot$ denotes Hadamard product, $\boldsymbol{W}^q$ and $\boldsymbol{W}^k$ denote learnable weights, $\sigma(\cdot)$ denotes a sigmoid function to scale the learnable weights into $[0, 1]$ for learning stability. We use two Gram matrices $\boldsymbol{F}^q \boldsymbol{F}^{qT}$ and $\boldsymbol{F}^k \boldsymbol{F}^{kT}$ derived from the self-correlation for modeling the feature domain [35], and hence effectively capture the context and target feature distributions. Moreover, we introduce learnable parameters $\boldsymbol{W}^q$ and $\boldsymbol{W}^k$ to be applied on the Gram matrices to dynamically highlight the most relevant context and target features, improving efficiency and avoiding overfitting as illustrated in [29]. The domain difference $\boldsymbol{A}^d$ captures the distinctions between context features and target features, which is applied on $\boldsymbol{F}^v$ at a later stage to highlight variations that are essential for distinguishing contextual dependencies from target-specific characteristics. To further capture finer-grained domain differences between context and target features, we introduce two auxiliary branches $\boldsymbol{A}^{td}$ and $\boldsymbol{A}^{cd}$,

$$\boldsymbol{A}^{td} = \boldsymbol{A}^d \odot \sigma(\boldsymbol{F}^q \boldsymbol{M}^t); \boldsymbol{A}^{cd} = \boldsymbol{A}^d \odot \sigma(\boldsymbol{F}^k \boldsymbol{M}^c), \tag{6}$$

where $\boldsymbol{M}^t \in \mathbb{R}^{C^h \times T}$ and $\boldsymbol{M}^c \in \mathbb{R}^{C^h \times T}$ are learnable matrices, which helps highlight the important domain-difference features relating to $\boldsymbol{F}^q$ and $\boldsymbol{F}^k$, respectively. $\sigma(\boldsymbol{F}^q \boldsymbol{M}^t)$ and $\sigma(\boldsymbol{F}^k \boldsymbol{M}^c)$ can be treated as gating matrices. As they depend on the input features, their dynamic nature helps the model to modulate its focus effectively to capture subtle relational patterns specific to each input which would be missed otherwise, as pointed out in [31]. Finally, the predicted features $\boldsymbol{F}^p$ are obtained as,

$$\boldsymbol{F}^p = \mathcal{F}_{\text{Linear}}((\underbrace{\sigma(\widehat{\boldsymbol{A}}^d)}_{dynamic} + \underbrace{\sigma(\widehat{\boldsymbol{A}}^{td})}_{dynamic} + \underbrace{\sigma(\widehat{\boldsymbol{A}}^{cd})}_{dynamic})\boldsymbol{F}^v). \tag{7}$$

The three terms are all dynamic and adaptable to the inputs so that the model could flexibly adapt to task-specific reasoning rules, thereby equipping it with enhanced reasoning capabilities compared to merely learning static mapping relationships [18, 19]. However, relying solely on dynamic parameters may lead to training instability, as the parameters continuously drift. We hence incorporate static linear and convolutional layers to ensure that the features are mapped into a relatively stable space.

### 3.5 Multi-granularity Rule Mixer

To better establish the reasoning rules, we propose the MRM to explore combinations of rules through multi-granular rule mixing, thereby enhancing the feature interaction. Specifically, denote the features after DDCP in the $h$-th view of layer $l$ as $\boldsymbol{F}_h^l$. To mix $\boldsymbol{F}_h^l$ on multiple levels of granularity, we first adopt a channel-wise mapping $\mathcal{F}_c^{l,h}$ and a token-wise mapping $\mathcal{F}_t^{l,h}$ in parallel at the fine-grained feature level to capture rule combinations across channels and tokens, respectively,

$$\boldsymbol{M}_h^l = \mathcal{F}_c^{l,h}(\boldsymbol{F}_h^l) + \mathcal{F}_t^{l,h}(\boldsymbol{F}_h^l). \tag{8}$$

We then apply a coarser row-wise mapping $\mathcal{F}_r^{l,h}$ and a sample-wise mapping $\mathcal{F}_s^{l,h}$ to further refine the rules, yielding an integrated feature map $\hat{\boldsymbol{V}}_h^l = \mathcal{F}_{\text{Conv}}(\mathcal{F}_s^{l,h}(\mathcal{F}_r^{l,h}(\boldsymbol{M}_h^l)))$, where the convolution $\mathcal{F}_{\text{Conv}}$ combines these relationships. By utilizing multi-granularity mapping, the model refines the complex rules from various perspectives and granularity and hence captures more rule combinations.

## 4 Experimental Results

### 4.1 Experiment Settings

DSRF is compared with 16 state-of-the-art models, WReN [7], CoPINet [36], SCL [37], SRAN [8], DCNet [38], MRNet [9], HCV-ARR [14], AlgeMR [13], ARII [39], PredRNet [15], STSN [40], SCAR [17], DRNet [16], TRIVR [41], HP²AI [19] and Slot Abstractors [42] on 6 RPM datasets, namely RAVEN [6], I-RAVEN [8], RAVEN-FAIR [9], PGM [7], Unicode Analogies (UA) [10] and RPM-like Video Prediction (RVP) [41]. Unless otherwise stated, results of compared methods are obtained using the source codes provided by the authors. We follow the standard evaluation protocol in [6, 8–10]. The input image size is $80 \times 80$. No other form of auxiliary supervision is incorporated during training. The Adam optimizer is applied with a learning rate of 1e-3 and weight decay of 1e-5. The batch size is set to 128. More details are provided in Appendix.

Table 1: Comparisons on RAVENs [6, 8, 9]. Other results are obtained from their original papers.

| Models | Avg. | O-RVN | I-RVN | RVN-F |
|---|---|---|---|---|
| MRNet (CVPR'21) [9] | 82.2 | 84.0 | 81.0 | 81.6 |
| HCV-ARR (AAAI'23) [14] | 92.2 | 87.3 | 93.9 | 95.4 |
| AlgeMR (CVPR'23) [13] | 93.5 | 92.9 | 93.2 | 94.3 |
| SCAR (AAAI'24) [17] | 93.8 | 92.8 | 94.7 | 93.9 |
| PredRNet (ICML'23) [15] | 96.5 | 95.8 | 96.5 | 97.1 |
| DRNet (AAAI'24) [16] | 97.4 | 96.9 | 97.6 | 97.6 |
| HP$^2$AI (ACM MM'24) [19] | 98.9 | 98.8 | 99.4 | 98.6 |
| *DSRF-S* (Ours) | 98.7 | 98.6 | 98.8 | 98.8 |
| *DSRF-M* (Ours) | 99.2 | 99.1 | 99.4 | 99.0 |
| *DSRF-L* (Ours) | **99.2** | **99.1** | **99.4** | **99.1** |

Table 2: Comparison on RAVENs under OOD settings [6, 8, 9].

| Models | Avg. | Train on O-RVN | | Train on I-RVN | | Train on RVN-F | | (G) | (M) |
|---|---|---|---|---|---|---|---|---|---|
| | | I-RVN | RVN-F | O-RVN | RVN-F | O-RVN | I-RVN | | |
| MRNet (CVPR'21) [9] | 75.7 | 65.8 | 81.2 | 76.5 | 78.6 | 78.8 | 73.4 | 3.58 | 4.96 |
| HCV-ARR (AAAI'23) [14] | 87.3 | 79.4 | 91.9 | 81.4 | 89.4 | 94.1 | 87.3 | 8.70 | 7.72 |
| SCAR (AAAI'24) [17] | 90.7 | 84.7 | 95.9 | 91.9 | 94.5 | 91.3 | 85.6 | 0.44 | 0.55 |
| HP$^2$AI (ACM MM'24) [19] | 92.3 | 87.7 | 96.9 | 89.9 | 94.0 | 95.3 | 90.0 | 12.82 | 6.38 |
| PredRNet (ICML'23) [15] | 94.3 | 89.2 | 97.9 | 93.9 | 97.3 | 96.2 | 91.4 | 2.05 | 1.28 |
| *DSRF-S* (Ours) | 96.7 | 95.1 | 98.5 | 95.9 | 98.2 | 97.3 | 95.4 | 10.10 | 11.64 |
| *DSRF-M* (Ours) | 97.3 | 95.6 | 98.7 | 97.4 | 98.7 | 97.8 | 95.8 | 13.92 | 23.86 |
| *DSRF-L* (Ours) | **97.7** | **95.9** | **98.9** | **97.9** | **98.9** | **98.1** | **96.5** | 22.10 | 42.29 |

## 4.2 Comparisons with State-of-the-Art Models

**Results on RAVEN Datasets.** Tab. 1 presents the results on the three RAVEN datasets [6, 8, 9]. We have following observations. 1) DSRF-S is comparable to the previous best performing model, HP$^2$AI [19], while DSRF-M and DSRF-L consistently outperform it across three datasets [6, 8, 9], with a gain of 0.3% over HP$^2$AI [19]. The performance gain is attributed to the scalability of DSRF that boosts reasoning ability and the dynamic designs that adapt to complex relations. 2) Furthermore, on RVN-F [9], DSRF-L achieves the largest performance gain over HP$^2$AI [19], reducing the error from 1.4% to 0.9%, highlighting the superiority of DSRF-L in widening the network to capture more complex rules. 3) Lastly, despite being a lighter version, DSRF-S attains a competitive average accuracy of 98.7% across three datasets [6, 8, 9]. Our DSRF-M and DSRF-L outperform it by 0.5%, contributing an enhanced reasoning ability, while showing no sign of overfitting.

We further conduct Out-Of-Distribution (OOD) experiments on the three RAVEN datasets [6, 8, 9], where models are trained on one dataset while evaluated on another. Tab. 2 summarizes the results. We can observe the following. 1) Our models consistently and significantly outperforms all compared methods across all settings, reaching average accuracies of 96.7%, 97.3%, and 97.7% for small, medium, and large models, respectively. Specifically, DSRF-L outperforms the second best model, PredRNet [15], by 3.4% on average. 2) When trained on O-RVN [6] and tested on I-RVN [8], all the compared models perform poorly with accuracies below 90%, showing the defects of static reasoning models in handing domain shift. In contrast, DSRF-L maintains an accuracy of 95.9%, achieving a performance gain of 6.7% compared to PredRNet [15]. 3) Across six OOD settings, the proposed models demonstrate minimal accuracy fluctuations, indicating that our models maintain their robustness across different scenarios.

As shown in Tab. 2, the DSRF-S requires competitive 10.10 GFLOPs with 11.64M parameters, while significantly outperforming other methods in accuracy. As the model scales up, despite the increase of GFLOPs and model parameters, the DSRF-M and DSRF-L consistently bring performance gain, indicating that the MVRP could well expand the reasoning network without overfitting.

Table 3: Comparison results on the Unicode Analogy/Unicode Analogy OOD [10].

| Models | Accuracy (%) on Unicode / Unicode-OOD | | | | | |
| --- | --- | --- | --- | --- | --- | --- |
| | **Avg.** | Arith | Const | Dist3 | Prog | Union |
| MRNet (CVPR'21) [9] | 24.3/22.9 | 25.5/20.9 | 22.9/23.7 | 24.4/26.1 | 27.4/23.5 | 21.5/20.2 |
| SCAR (AAAI'24) [17] | 31.4/23.1 | 26.6/22.3 | 34.0/25.0 | 28.6/24.9 | 25.9/24.5 | 41.8/18.9 |
| HCV-ARR (AAAI'23) [14] | 25.9/24.1 | 26.8/23.6 | 25.0/26.4 | 23.5/26.4 | 24.2/23.0 | 30.0/26.6 |
| PredRNet (ICML'23) [15] | 57.7/27.7 | 65.5/26.3 | 51.4/30.7 | 48.5/32.4 | 57.6/27.0 | 65.6/22.3 |
| HP²AI (ACM MM'24) [19] | 55.2/30.2 | 59.4/26.2 | 50.6/28.8 | 53.7/34.6 | 54.2/28.4 | 58.2/33.0 |
| *DSRF-S* (Ours) | 72.1/34.5 | 74.4/29.1 | 61.0/34.3 | 76.6/37.9 | 72.8/33.0 | 75.6/38.2 |
| *DSRF-M* (Ours) | 73.2/35.9 | 76.5/30.3 | 61.9/35.6 | 76.8/38.9 | 73.6/35.4 | 77.1/39.5 |
| *DSRF-L* (Ours) | **73.8/37.3** | **79.4/32.5** | **61.9/36.4** | **77.0/39.8** | **73.7/36.7** | **77.2/41.2** |

Table 4: Comparisons on PGM [7]. Results of other methods are obtained from their original papers.

| Models | Avg. | PGM-N | PGM-I | PGM-E | HAP | HTP | HT | HALT | HASC |
| --- | --- | --- | --- | --- | --- | --- | --- | --- | --- |
| CoPINet (NIPS'19) [36] | - | 56.4 | 51.2 | 16.4 | - | - | - | - | - |
| WReN (ICML'18) [7] | 32.4 | 62.6 | 64.4 | 17.2 | 27.2 | 41.9 | 19.0 | 14.4 | 12.5 |
| DCNet (ICLR'21) [38] | - | 68.6 | 59.7 | 17.8 | - | - | - | - | - |
| SRAN (AAAI'21) [8] | - | 71.3 | 60.1 | 18.4 | - | - | - | - | - |
| MRNet (CVPR'21) [9] | 43.4 | 93.4 | 68.1 | 19.2 | 38.4 | 55.3 | 25.9 | **30.1** | 16.9 |
| ARII (NIPS'22) [39] | 45.5 | 88.0 | 72.0 | 29.0 | 50.0 | 64.1 | 32.1 | 16.0 | 12.7 |
| PredRNet (ICML'23) [15] | 47.8 | 97.4 | 70.5 | 19.7 | 63.4 | 67.8 | 23.4 | 27.3 | 13.1 |
| STSN (ICLR'23) [40] | - | 98.2 | 78.5 | 20.4 | - | - | - | - | - |
| HP²AI (ACM MM'24) [19] | - | 99.3 | 80.0 | 22.6 | - | - | - | - | - |
| Slot Abs.(ICML'24) [42] | 51.9 | 91.5 | **91.6** | **39.3** | 63.3 | 78.3 | 20.4 | 16.7 | 14.3 |
| *DSRF-S* (Ours) | 50.9 | 99.3 | 82.4 | 25.2 | 63.1 | 75.7 | 29.3 | 16.2 | 16.1 |
| *DSRF-M* (Ours) | 52.7 | 99.5 | 84.5 | 27.4 | 65.7 | 78.8 | 31.3 | 17.4 | 16.7 |
| *DSRF-L* (Ours) | **54.9** | **99.8** | 87.3 | 31.5 | **67.7** | **82.3** | **34.2** | 18.9 | **17.8** |

**Results on Unicode Dataset.** We conduct experiments on the UA dataset [10] across five rule types under two settings: 1) Normal setting, where the dataset is randomly divided into training, validation, and test sets; and 2) OOD setting, where all the symbols in the training set do not appear in the test set. The UA dataset [10] provides a comprehensive evaluation of conceptual schema across multiple levels of abstraction. The following observations can be made from Tab. 3: 1) All three proposed DSRF models significantly and consistently outperform existing methods across five rule types under both settings. Specifically, in the normal setting, the proposed DSRF-L achieves the highest average accuracy of 73.8%, demonstrating a substantial performance gain of 16.1% compared to the previous-best model, PredRNet [15]. 2) Under the OOD setting, all models experience a significant performance drop due to distribution shifts and unfamiliar data patterns. While most existing models achieve poor accuracies of approximately 25%, approaching random guessing, our DSRF-L obtains a significantly higher average accuracy of 37.3%, presenting a notable improvement of 7.1% over the previous-best model HP²AI [19], highlighting the robustness of our model by dynamically capturing the domain difference.

**Results on PGM Dataset.** Then, we benchmark our DSRF models against state-of-the-art models on the PGM dataset [7]. Following [42], experiments are evaluated across eight regimes: neutral (PGM-N), interpolation (PGM-I), extrapolation (PGM-E), held-out attribute pairs (HAP), held-out triple pairs (HTP), held-out triples (HT), held-out attribute line type (HALT), held-out attribute shape color (HASC). The last seven regimes evaluate OOD generalization. (See Appendix B.1.4 for more details.) The results in Tab. 4 reveal the following: 1) Both DSRF-M and DSRF-L surpass existing approaches across nearly all regimes. Notably, DSRF-L achieves a new state-of-the-art average accuracy of 54.9%, substantially outperforming the previous best result of 51.9% by Slot Abstractors [42], validating our framework's effectiveness. 2) The performance gains of DSRF become even more pronounced under challenging held-out and transfer regimes such as HAP, HTP, HT and HASC, where DSRF-L reaches 82.3% on HTP and 34.2% on HT, significantly outperforming

Table 5: Comparisons on RVP dataset [41]. Other results are obtained from their original papers.

| CoPINet [36] | SRAN [8] | SCL [37] | TRIVR [41] | **DSRF-S** | **DSRF-M** | **DSRF-L** |
|---|---|---|---|---|---|---|
| 62.3 | 63.9 | 66.3 | 71.6 | **85.4** | **87.7** | **88.8** |

Table 6: Ablation of major components of DSRF-M on three RAVEN datasets [6, 8, 9].

| | **Avg.** | O-RVN | I-RVN | RVN-F |
|---|---|---|---|---|
| w/o DDCP | 95.2 | 94.9 | 95.1 | 95.6 |
| w/o GARB | 95.9 | 95.9 | 95.6 | 96.3 |
| w/o CNN gate | 98.0 | 97.9 | 97.8 | 98.3 |
| w/o dynamic gate | 98.0 | 97.8 | 98.1 | 98.2 |
| w/o MRM | 98.7 | 98.7 | 98.8 | 98.6 |
| *DSRF-M* | **99.2** | **99.1** | **99.4** | **99.0** |

all baselines and demonstrating strong domain-difference modeling and compositional generalization. 3) We observe a clear monotonic improvement as we scale the model from DSRF-S to DSRF-M and then to DSRF-L. This is evidenced across all metrics: the overall average accuracy sees a steady gain from 50.9% to 52.7% and finally to 54.9%. The improvement is especially notable in the HTP regime, where performance increases from 75.7% (DSRF-S) to 78.8% (DSRF-M) and reaches 82.3% (DSRF-L). This consistent progression demonstrates that our framework's effectiveness scales reliably with increased model capacity.

**Results on RVP Dataset.** Lastly, we evaluate DSRF variants on the RVP dataset [41] for real-world traffic scenarios. RVP is built on the RPM framework, where future frames are predicted from two historical frames sampled every 15 frames, leveraging reasoning patterns learned from other videos. As shown in Tab. 5, our model largely outperforms the second-best model, thanks to the MVRP that enables the integration of more reasoning blocks into the reasoning module, allowing the model to capture more complex and subtle underlying rules in real-world scenarios. In contrast, previous models with simpler reasoning modules struggle to generalize to such complex environments.

### 4.3 Ablation Studies

**Ablation of Major Components.** To evaluate the major components of DSRF, we conduct an ablation study across three RAVEN datasets [6, 8, 9], by systematically removing one component at a time to assess its individual performance contribution. The results are presented in Tab. 6. When removing DDCP and GARB, a single convolutional layer is used to replace them. It can be observed that the accuracy on all three datasets will drop significantly if removing either DDCP or GARB, *i.e.*, 4.0% for DDCP and 3.3% for GARB, demonstrating their importance in dynamically encapsulating the task-specific rules. Additionally, removing the CNN gate in DDCP or removing the dynamic gate in GARB both leads to a significant accuracy decline of 1.2%, highlighting their significance in dynamically emphasizing the relevant features while suppressing irrelevant features. Finally, MRM also shows its effectiveness in aggregating features from different levels of granularity.

**DSRF as Plug-and-Play Reasoning Module.** The proposed DSRF could serve as a plug-and-play reasoning module. We replace the original reasoning module of state-of-the-art models, *i.e.*, RM+PM of MRNet [9], ARR of HCV-ARR [14], PRB of PredRNet [15] and PredAI of HP$^2$AI [19], by DSRF-S and DSRF-M respectively, and conduct comparison experiments on the three RAVEN datasets [6, 8, 9] as shown in Tab. 7. We can observe that DSRF-S and DSRF-M consistently improve performance compared to the original reasoning modules, indicating their robustness and effectiveness in reasoning complex underlying relations. Lastly, DSRF-M consistently outperforms DSRF-S across all four perception modules on all three datasets, indicating the effectiveness of our scalable MVRP in boosting the model's reasoning ability.

**Lightweight DSRF.** The MVRP module enables scalable model efficiency by reducing the channel dimensions of feature mappings per view. This allows the creation of lightweight variants (DSRF-S-Light, DSRF-M-Light) that preserve core reasoning capabilities with significantly reduced parameters

Table 7: Replacing reasoning modules by DSRF-S and DSRF-M on RAVEN datasets [6, 8, 9].

| Perception | Reasoning | **Avg.** | O-RVN | I-RVN | RVN-F |
|---|---|---|---|---|---|
| MSE [9] | RM+PM [9] | 83.9 | 84.0 | 81.0 | 86.8 |
| | *DSRF-S* | 96.5 | 96.3 | 96.7 | 96.5 |
| | *DSRF-M* | **97.1** | **96.4** | **96.9** | **97.9** |
| HCV [14] | ARR [14] | 92.2 | 87.3 | 93.9 | 95.4 |
| | *DSRF-S* | 96.9 | 96.4 | 96.5 | 97.9 |
| | *DSRF-M* | **97.6** | **97.5** | **97.2** | **98.1** |
| RN-4B [15] | PRB [15] | 96.5 | 95.8 | 96.5 | 97.1 |
| | *DSRF-S* | 98.7 | 98.6 | 98.8 | 98.8 |
| | *DSRF-M* | **99.2** | **99.1** | **99.4** | **99.0** |
| HPALC [19] | PredAI [19] | 98.9 | 98.8 | 99.4 | 98.6 |
| | *DSRF-S* | 99.2 | 99.1 | 99.4 | 99.2 |
| | *DSRF-M* | **99.4** | **99.3** | **99.5** | **99.4** |

Table 8: Ablation of lightweight DSRF-S and DSRF-M on RAVENs [6, 8, 9] and PGM [7].

| | **Avg.** | O-RVN | I-RVN | RVN-F | PGM-N | PGM-I | PGM-E | **(G)** | **(M)** |
|---|---|---|---|---|---|---|---|---|---|
| DSRF-S | 83.8 | 98.7 | 98.6 | 98.8 | 99.3 | 82.4 | 25.2 | 10.10 | 11.64 |
| DSRF-S-Light | 83.5 | 98.5 | 98.5 | 98.6 | 99.1 | 81.7 | 24.8 | 4.06 | 4.96 |
| DSRF-M | 84.8 | 99.1 | 99.4 | 99.0 | 99.5 | 84.5 | 27.4 | 13.92 | 23.86 |
| DSRF-M-Light | 84.2 | 98.8 | 98.9 | 98.8 | 99.3 | 83.1 | 26.1 | 5.11 | 8.72 |

Table 9: Ablation of column-wise reasoning DSRF-M and DSRF-L on the PGM dataset [7].

| | **Avg.** | PGM-N | PGM-I | PGM-E | HAP | HTP | HT | HALT | HASC |
|---|---|---|---|---|---|---|---|---|---|
| DSRF-M | 52.7 | 99.5 | 84.5 | 27.4 | 65.7 | 78.8 | 31.3 | 17.4 | 16.7 |
| DSRF-M-RC | 55.8 | 99.7 | 88.9 | 35.1 | 68.9 | 82.1 | 34.1 | 19.1 | 18.1 |
| DSRF-L | 54.9 | 99.8 | 87.3 | 31.5 | 67.7 | 82.3 | 34.2 | 18.9 | 17.8 |
| DSRF-L-RC | 58.4 | 99.9 | 92.1 | 40.2 | 71.3 | 86.7 | 37.3 | 19.9 | 19.4 |

and GFLOPs. Evaluations on three RAVEN and PGM datasets (Tab. 8) confirm that these variants maintain competitive accuracy while achieving substantial efficiency gains.

**Column-wise reasoning of DSRF.** While DSRF's row-wise design efficiently handles standard row-centric RPMs, it faces limitations on datasets like PGM that emphasize column-wise reasoning. Although weight-sharing across rows allows for implicit column-wise pattern recognition, we explicitly enhance this capability by integrating a dedicated column-wise DDCP branch. The modularity of the DDCP design permits this seamless extension, enabling explicit fusion of column-wise features with the original row-wise processing. As shown in Tab. 9, the revised models (DSRF-M-RC and DSRF-L-RC) yield significant gains, improving the average reasoning accuracy of DSRF-M from 52.7% to 55.8% and DSRF-L from 54.9% to 58.4%, respectively.

## 5 Conclusion

The proposed Dynamic and Scalable Reasoning Framework is designed to tackle the challenges of scalability and the dynamic nature of abstract visual reasoning, in order to boost the reasoning ability of our model and generalize it well to novel tasks. Specifically, the proposed MVRP tackles the scalability challenges by hierarchically widening the network to capture a broad range of complex rules. In addition, the proposed DDCP blocks enhance the adaptability to diverse rules by dynamically highlighting task-specific relationships respective to the domain difference through a novel Gated Attention Reasoning Block. Extensive results on six AVR datasets demonstrate that DSRF outperforms state-of-the-art models on different datasets and tasks.

## Acknowledgment

This work was supported by Ningbo Science & Technology Bureau under Grant 2023Z138, 2023Z237, 2024Z110 and 2024Z124.

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

# A    Significance of This Research

Abstract Visual Reasoning (AVR) is a pivotal yet challenging research in artificial intelligence, focusing on inferring complex abstract rules from visual data [1]. Progress in this field is critical for enhancing machine comprehension and learning capabilities [1]. However, recent works [3, 4] indicate that LLMs often fail to comprehend abstract concepts and relationships within images for visual reasoning tasks. Additionally, although multimodal models attempt to integrate visual and linguistic information, they still fall short in complex visual reasoning tasks [43]. Specifically, as stated in [44], in AVR tasks, enhancing the amount of information is more important than enforcing disentangled representations because the amount of information has a greater impact on downstream task performance.

The proposed method provides a novel solution to tackle the challenges of existing AVR models on scalability and generalization ability. Specifically, the proposed method leverages the Multi-View Reasoning Pyramid to map high-level visual features into different views, and hierarchically constructs the complex reasoning rules through the layered pyramid, thereby greatly enhancing the scalability of the model. In addition, in contrast to the static reasoning networks in existing models, the proposed Dynamic Domain-Contrast Prediction Block incorporates different mechanisms to dynamically model the underlying rules, including a CNN gate to selectively highlight the important reasoning features, and a set of Gated Attention Reasoning Blocks to dynamically reason over a large number of rules. Furthermore, in each GARB, we model the distribution of context features and target features by utilizing Gram matrix, contrast the domain differences through learning parameters, and further modulate its focus to capture subtle relational patterns specific to the inputs by using two gate matrices. As shown in the experiments, our work pushes the edge of abstract visual reasoning, achieving state-of-the-art performance across six benchmark datasets.

# B    Experimental Settings

## B.1    Dataset Description

Table 10: Summary of benchmark datasets.

| Datasets | #Samples | #Images | #Attributes | #Relations |
|---|---|---|---|---|
| O-RVN [6] | 70K | 1.12M | 7 | 4 |
| I-RVN [8] | 70K | 1.12M | 7 | 4 |
| RVN-F [9] | 70K | 1.12M | 7 | 4 |
| PGM [7] | 1.42M | 22.72M | 5 | 5 |
| UA [10] | 50K | 0.45M | / | 5 |
| RVP [41] | 3K | 48K | / | / |

### B.1.1    Original RAVEN Dataset

The original RAVEN dataset [6] comprises 70,000 question sets, generated by converting sentences derived from an Attributed Stochastic Image Grammar (A-SIG) [45]. Each question set consists of eight context images and eight candidate options, resulting in a total of 1.12 million images. Each question is defined by five rule-governing attributes such as `Number`, `Position`, `Type`, `Size` and `Color`, and two noise attributes such as `Uniformity` and `Orientation`. Each attribute follows one of the four rules, namely `Constant`, `Progression`, `Arithmetic`, and `Distribute_Three`. This design yields an average of 6.29 rules per problem, which is significantly higher than the rule complexity in the PGM dataset [7]. The RAVEN dataset [6] is evenly divided into seven configurations, *i.e.*, Center, 2×2Grid (2×2G), 3×3Grid (3×3G), Left-Right (L-R), Up-Down (U-D), Out-InCenter (O-IC), and Out-InGrid (O-IG), which are generated using three levels from A-SIG [45]. The standard 10-fold evaluation protocol [6] is applied, where six folds are used for training, and two folds each for validation and testing. However, as identified in [8, 9], there is a loophole in the RAVEN dataset, *i.e.*, candidate options in the RAVEN dataset are created by modifying one single attribute from the correct answer, which allows models to easily select the correct answer by identifying common properties among candidates rather than understanding the underlying relational patterns in the context matrix. The samples of RAVEN are shown in Fig. 4.

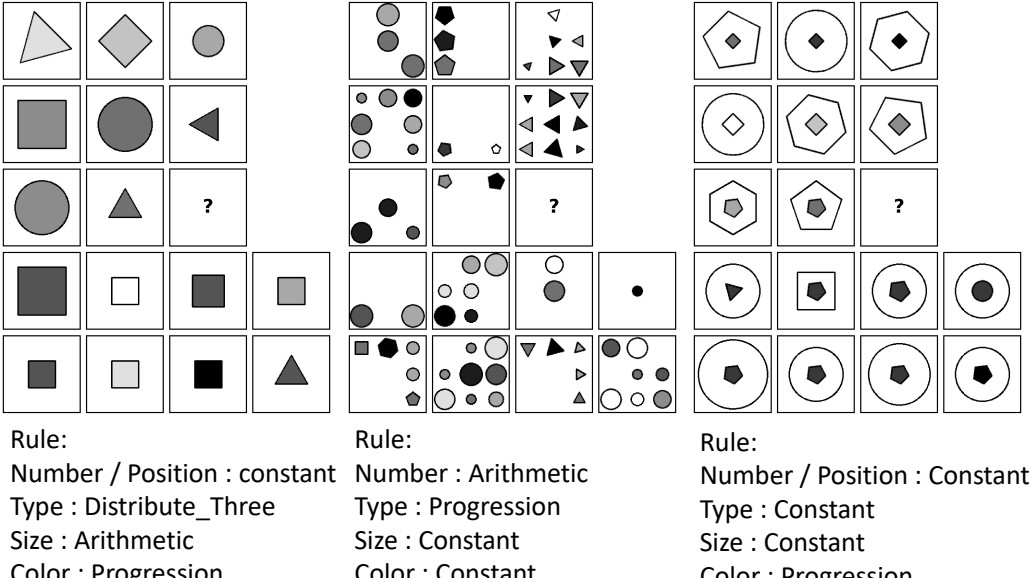

Rule:
Number / Position : constant
Type : Distribute_Three
Size : Arithmetic
Color : Progression

Rule:
Number : Arithmetic
Type : Progression
Size : Constant
Color : Constant

Rule:
Number / Position : Constant
Type : Constant
Size : Constant
Color : Progression

Figure 4: Three typical samples in RAVEN dataset.

### B.1.2 I-RAVEN Dataset

The I-RAVEN dataset [8] eliminates the vulnerability of the original RAVEN dataset [6], differing in the way of generating negative candidate answers. Specifically, it utilizes an Attribute Bisection Tree to generate impartial and balanced candidate options, where each option has exactly three differing attributes from the other through three iterations. This makes the candidate options more indistinguishable without identifying the context matrix, leading to a more rigorous and fair evaluation of the models' abstract reasoning capability. Other settings are exactly the same as the original RAVEN dataset [6]. The samples of I-RAVEN are shown in Fig. 5.

### B.1.3 RAVEN-FAIR Dataset

The RAVEN-FAIR dataset [9] also addresses the shortcut issue of the RAVEN dataset [6]. The negative candidates are systematically generated by randomly selecting an existing option, either a negative answer or the correct answer, and modifying one of its attributes. This design ensures that the correct answer cannot be selected by only identifying the most common candidate options. Other settings are exactly the same as the original RAVEN dataset [6]. Three samples of RAVEN-FAIR are shown in Fig. 6.

### B.1.4 PGM Dataset

The PGM dataset [7] is widely recognized as the first large-scale RPM benchmark to evaluate the reasoning ability of deep learning models. It consists of 1.42M question sets, with 1.2M questions for training, 20K for validation, and 200K for testing. Each question set contains 8 context images and 8 candidate images, resulting in a total of 22.72M images. Each matrix in PGM [7] is a set of triples, $\mathcal{S} = \{[r, o, a] | r \in \mathcal{R}, o \in \mathcal{O}, a \in \mathcal{A}\}$, where $\mathcal{R} = \{\texttt{Progression}, \texttt{XOR}, \texttt{OR}, \texttt{AND}, \texttt{Consistent\_Union}\}$ defines the set of rules, $\mathcal{O} = \{\texttt{Shape}, \texttt{Line}\}$ represents the set of objects and $\mathcal{A} = \{\texttt{Size}, \texttt{Type}, \texttt{Color}, \texttt{Position}, \texttt{Number}\}$ indicates the set of attributes. The PGM dataset comprises 8 regimes.

**Neutral** regime (PGM-N): In this regime, both training and test sets can contain any relation–object–attribute triples. They are disjoint at the pixel level but share identical structural distributions, serving as the baseline for reasoning performance.

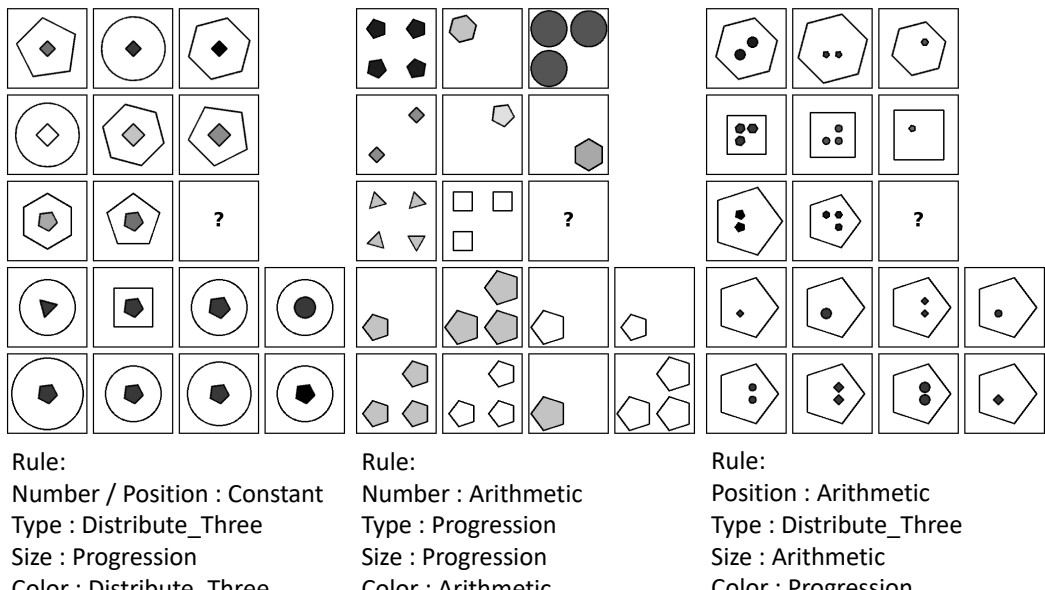

Rule:
Number / Position : Constant
Type : Distribute_Three
Size : Progression
Color : Distribute_Three

Rule:
Number : Arithmetic
Type : Progression
Size : Progression
Color : Arithmetic

Rule:
Position : Arithmetic
Type : Distribute_Three
Size : Arithmetic
Color : Progression

Figure 5: Three typical samples in I-RAVEN dataset.

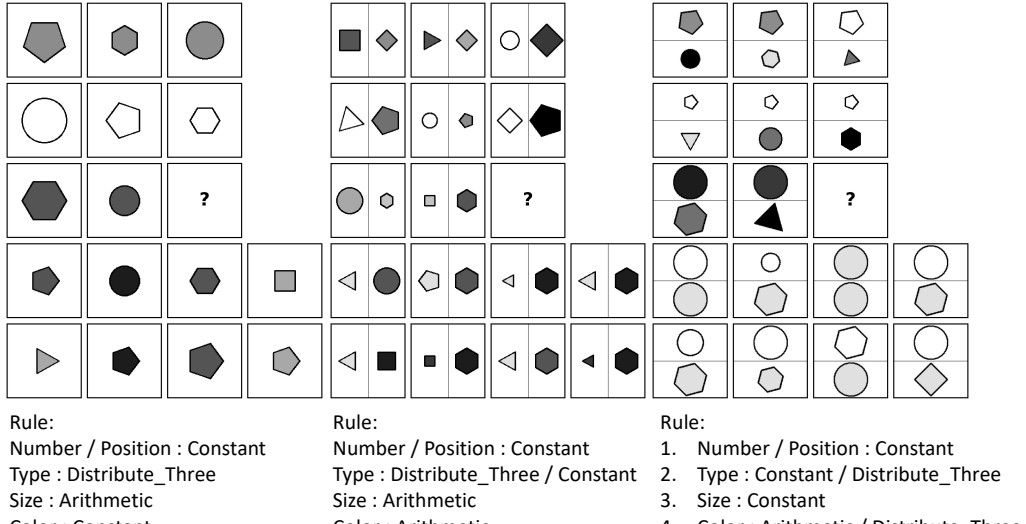

Rule:
Number / Position : Constant
Type : Distribute_Three
Size : Arithmetic
Color : Constant

Rule:
Number / Position : Constant
Type : Distribute_Three / Constant
Size : Arithmetic
Color : Arithmetic

Rule:
1. Number / Position : Constant
2. Type : Constant / Distribute_Three
3. Size : Constant
4. Color : Arithmetic / Distribute_Three

Figure 6: Three typical samples in RAVEN-FAIR dataset.

**Interpolation** regime (PGM-I): Training and test sets share the same attribute types, but ordered attributes like color and size use even-indexed values in training and odd-indexed values in testing. It evaluates the model's ability to interpolate within known attribute ranges.

**Extrapolation** regime (PGM-E): Training samples include the lower half of ordered attribute values, while testing uses the upper half. This regime measures whether the model can extrapolate to unseen attribute ranges beyond the training distribution.

**Held-out Attribute Pairs** regime (HAP): In this regime, 20 viable attribute pairs are defined, with 16 used for training and 4 held out for testing. While training samples contain each attribute separately,

test samples include both together, assessing compositional generalization to unseen attribute co-occurrences.

**Held-out Triple Pairs** regime (HTP): In this regime, each structure contains at least two triples, forming 400 viable triple pairs, with 360 used for training and 40 reserved for testing. The held-out pairs never co-occur in training, evaluating the model's ability to generalize to unseen combinations of triple relations.

**Held-out Triples** regime (HT): In this regime, 7 of the 29 unique triples are randomly reserved for testing, ensuring all attribute types are represented. These triples never appear in training, and every test sample includes at least one, evaluating reasoning over entirely unseen combinations.

**Held-out Attribute Line Type** regime (HALT): In this regime, all line–type combinations are excluded from the training data but appear in every test sample. It tests the model's structural generalization to unseen line-style and type relations.

**Held-out Attribute Shape Color** regime (HASC): In this regime, no shape–color combinations are included in the training data, but every test sample contains at least one. It evaluates the model's ability to generalize compositionally to unseen pairings of object shape and color.

Most methods are evaluated on the *Neutral* regime, where the training and test sets are sampled from the same distribution of the *Neutral* regime. Other regimes are designed to assess the model's generalization capability, where the test set contains rules that are excluded from the training and validation sets, leading to a rigorous evaluation under Out-Of-Distribution (OOD) settings. The samples in PGM are shown in Fig. 7.

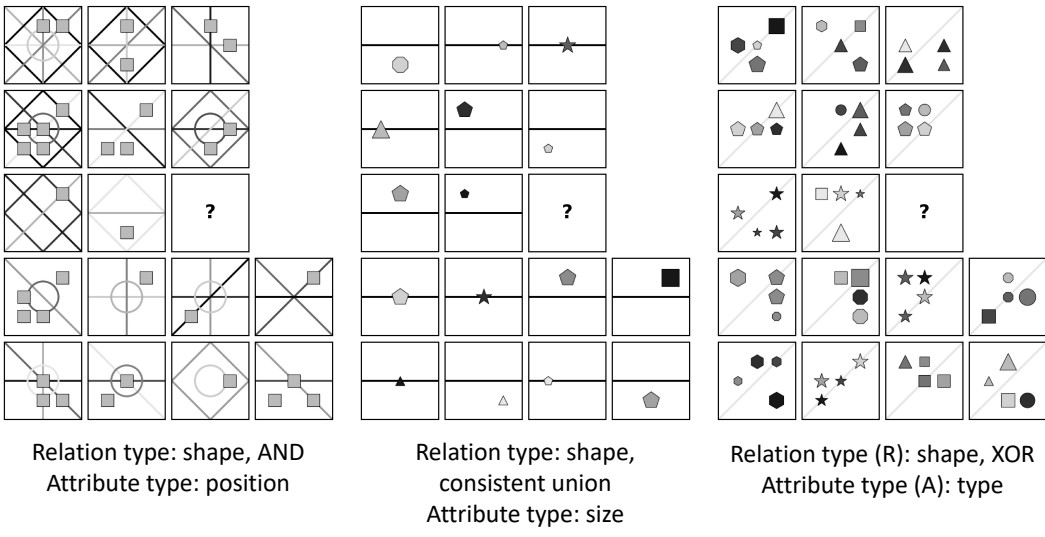

Relation type: shape, AND
Attribute type: position

Relation type: shape,
consistent union
Attribute type: size

Relation type (R): shape, XOR
Attribute type (A): type

Figure 7: Three typical samples in PGM dataset.

### B.1.5 Unicode Analogies Dataset

The Unicode Analogies (UA) dataset [10] provides a comprehensive framework for representing conceptual schema across multiple levels of abstraction. Each problem is generated by one of the five rule types, *i.e.* `Constant`, `Progression`, `Arithmetic`, `Distribution_Three`, and `Union`. The UA dataset [10] contains 2500 annotated characters with an average of 2.8 annotated features per character. Following the standard evaluation protocol in [10], a 10-fold evaluation is used, with seven folds for training, and one fold for validation and two folds for testing. In addition, it introduces complexities by blurring the lines between objects and features, as well as between perception and cognition. Compared to the three RAVEN datasets [6, 8, 9], the UA dataset [10] poses additional challenges as it requires models to incorporate contextual information at all stages of problem solving [10]. The samples in Unicode are shown in Fig. 8.

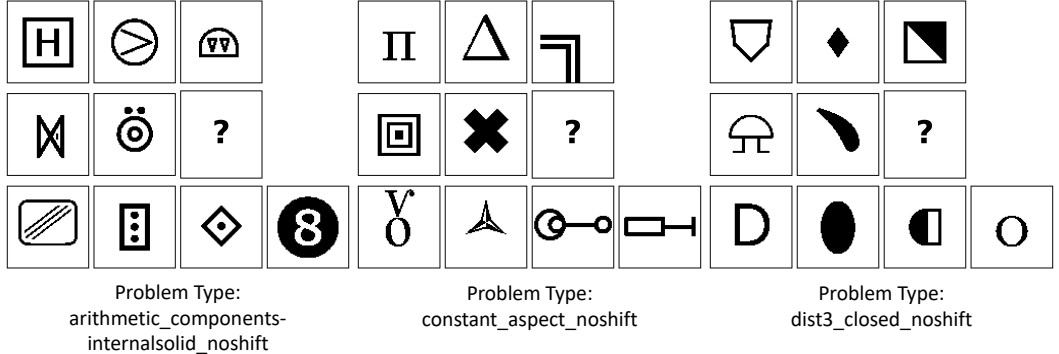

Problem Type:
arithmetic_components-
internalsolid_noshift

Problem Type:
constant_aspect_noshift

Problem Type:
dist3_closed_noshift

Figure 8: Three typical samples in the Unicode dataset.

### B.1.6 RVP Dataset

The RVP dataset [41] is built upon the UA-DETRAC dataset [46], which includes 100 challenging real-world traffic videos captured from 24 locations using a Canon EOS 550D camera. These videos represent diverse traffic conditions such as urban highways, intersections, and T-junctions. The original UA-DETRAC dataset contains over 140,000 annotated frames with details like vehicle type, illumination, occlusion, truncation ratio, and bounding boxes. Under the RPM framework, RVP predicts a future frame based on two historical frames sampled every 15 frames, using reasoning rules derived from other videos. Unlike traditional video prediction tasks, RVP uses non-consecutive frames, making the task more challenging. The goal is to select the correct answer from eight candidate frames by analyzing the progression in vehicle size and position. The incorrect options are sampled at least 15 frames apart from the correct answer to avoid trivial clues. In total, the dataset includes 3,000 questions and 48,000 frames, featuring visually complex scenes with multiple vehicles and detailed urban backgrounds. The samples in RVP are shown in Fig. 9.

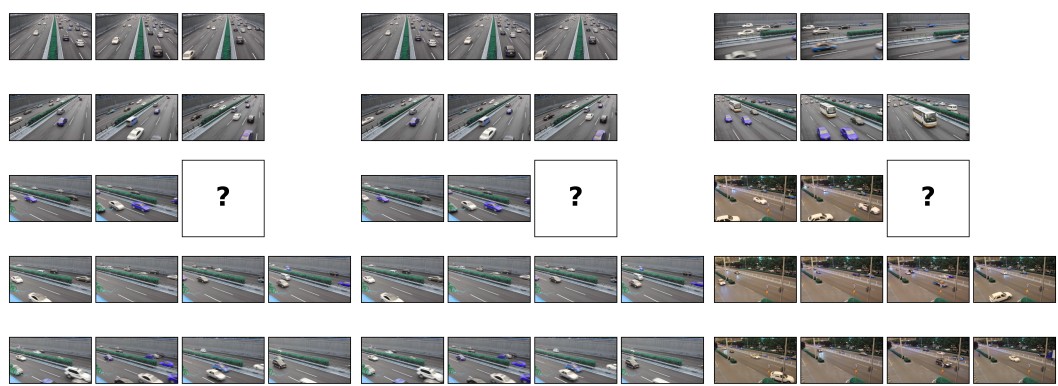

Figure 9: Three typical samples in RVP dataset.

### B.2 Detailed Description of Compared Methods

The proposed DSRF is compared with the following state-of-the-art methods.

**WReN** [7] utilizes a Relation Network to derive the inter-feature relations for solving RPM problems.

**CoPINet** [36] estimates the probability of each candidate answer by applying a contrastive module on top of a perception module built from ResNet blocks, using the question panel and each candidate as the input.

**SCL** [37] is designed to discover the underlying compositional structure. It consists of three integrated networks: an object network, an attribute network, and a relationship network.

**SRAN** [8] employs a hierarchical rule embedding module and a gated embedding fusion module to produce rule embeddings from two row sequences.

**DCNet** [38] incorporates a rule contrast module and a choice contrast module to leverage the intrinsic structure of RPMs, enhancing distinctions among options by comparing latent rules across rows.

**MRNet** [9] utilizes a multi-resolution convolution layer for visual perception and computes the row similarity as the reasoning strategy.

**ARII** [39] is a framework that learns abstract rule representations through internal inference mechanisms, combining a rule encoder, a reasoner, and an internal referrer. It repeatedly applies the same rule to different instances to achieve a comprehensive understanding.

**HCV-ARR** [14] utilizes a mixed model combining convolutional blocks and vision transformer blocks to capture multi-level features from RPM images, and employs an attention mechanism to dynamically determine the relations between a row/column of panel images.

**AlgeMR** [13] includes an object detector to identify discrete entity attributes and uses algebraic methods such as Gröbner bases and ideal containment to solve RPMs as computational problems.

**SCAR** [17] utilizes a Structure-Aware dynamic Layer (SAL) that enables the processing of AVR tasks with diverse structures by adapting its weights to the problem instance.

**PredRNet** [15] employs sequential residual convolutional layers to extract high-level visual features from images and utilizes convolutional blocks to identify abstract rules by predicting the target images using the context images.

**STSN** [40] combines slot attention for object-centric encoding and a transformer for reasoning, scoring each answer by concatenating its slots with context panel slots, and optimizing with both task and reconstruction losses.

**DRNet** [16] is a dual-stream neural network inspired by the two-stream visual processing hypothesis, which achieves strong generalization on RPM benchmarks by integrating spatial and semantic features to extract abstract reasoning rules.

**TRIVR** [41] is a two-stage visual reasoning model that separates perception and reasoning to reflect better the human approach to solve RPM problems. By introducing a "2+1" formulation to extract explicit reasoning rules from each sample, TRIVR significantly reduces the model complexity and outperforms state-of-the-art methods on multiple RPM-like datasets.

**HP$^2$AI** [19] extracts multi-scale visual features via a hierarchical encoder and infers row-wise relations using Predictive Analogy-Inference (PredAI) blocks, focusing on key attributes to solve RPM tasks effectively.

**Slot Abstractors** [42] integrates slot-based object-centric representations with the scalable, multi-head Transformer architecture. By incorporating strong relational inductive biases, it effectively handles reasoning tasks involving many objects and multiple relations.

### B.3 Implementation Details

We strictly follow the standard evaluation protocol outlined in [6, 8–10, 40]. Input images are resized to $80 \times 80$, and datasets are divided into training, validation, and test sets, with the validation set used for hyper-parameter tuning. No additional auxiliary supervision is employed during training. We apply Muon to hidden-layer parameters with dimension at least 2 using a learning rate of 3e-3, and Adam to all remaining parameters using a learning rate of 1e-3, with weight decay set to 1e-5. The models are trained with a batch size of 128 on an Intel Xeon Silver 4216 CPU with two NVIDIA RTX A5000 GPUs. The code will be released upon the acceptance of this paper.

Table 11: Ablation studies of different reasoning blocks on RAVEN [6], I-RAVEN [8], and RAVEN-FAIR [9]. The proposed GARB significantly improves the reasoning accuracy compared with other reasoning modules.

| Datasets | Components | Accuracy (%) on Different RAVEN Configurations | | | | | | | |
|---|---|---|---|---|---|---|---|---|---|
| | | Avg. | Center | 2×2G | 3×3G | L-R | U-D | O-IC | O-IG |
| O-RVN [6] | PredAI | 97.5 | 99.9 | 96.7 | 92.6 | 99.9 | 99.7 | 99.7 | 93.8 |
| | PRB | 98.2 | 100.0 | 98.1 | 94.5 | 98.2 | 99.8 | 99.7 | 97.4 |
| | DDCP | **99.1** | **100.0** | **99.3** | **96.2** | **99.9** | **100.0** | **99.9** | **98.2** |
| I-RVN [8] | PredAI | 97.3 | 99.9 | 96.9 | 91.8 | 99.9 | 99.9 | 99.7 | 92.9 |
| | PRB | 97.9 | 99.9 | 98.1 | 93.5 | 99.7 | 99.9 | 99.9 | 94.5 |
| | DDCP | **99.4** | **100.0** | **99.9** | **97.6** | **100.0** | **99.9** | **100.0** | **98.4** |
| RVN-F [9] | PredAI | 98.2 | 99.9 | 98.1 | 96.5 | 99.9 | 100.0 | 100.0 | 93.1 |
| | PRB | 98.5 | 100.0 | 98.7 | **97.4** | 99.9 | **100.0** | 99.9 | 93.8 |
| | DDCP | **99.0** | **100.0** | **99.8** | 97.3 | **100.0** | 99.9 | **100.0** | **96.3** |

Table 12: Ablation studies of MRM on RAVEN [6], I-RAVEN [8], and RAVEN-FAIR [9] datasets.

| | Components | Accuracy (%) on Different RAVEN Configurations | | | | | | | |
|---|---|---|---|---|---|---|---|---|---|
| | | Avg. | Center | 2×2G | 3×3G | L-R | U-D | O-IC | O-IG |
| O-RVN [6] | MLP | 98.7 | 99.9 | 98.4 | 96.1 | 99.9 | 99.9 | 99.9 | 96.9 |
| | CNN | 98.9 | 100.0 | 99.3 | 95.8 | **100.0** | 100.0 | 99.9 | 97.1 |
| | Attention | 98.5 | 100.0 | 98.4 | 95.1 | 99.9 | 100.0 | 99.9 | 96.4 |
| | MRM | **99.1** | **100.0** | **99.3** | **96.2** | 99.9 | **100.0** | **99.9** | **98.2** |
| I-RVN [8] | MLP | 98.9 | 100.0 | 99.3 | 95.9 | 99.9 | 100.0 | 99.9 | 97.2 |
| | CNN | 99.1 | 100.0 | 99.5 | 96.4 | 100.0 | **100.0** | 100.0 | 97.9 |
| | Attention | 98.6 | 100.0 | 98.5 | 95.1 | 99.9 | 99.9 | 99.8 | 96.7 |
| | MRM | **99.4** | **100.0** | **99.9** | **97.6** | **100.0** | 99.9 | **100.0** | **98.4** |
| RVN-F [9] | MLP | 98.5 | 100.0 | 98.9 | 97.2 | 99.9 | **100.0** | 99.9 | 93.9 |
| | CNN | 98.7 | 100.0 | 99.2 | **97.8** | 99.9 | 99.9 | 100.0 | 94.3 |
| | Attention | 98.3 | 99.9 | 98.5 | 96.6 | 100.0 | 99.9 | 99.8 | 93.4 |
| | MRM | **99.0** | **100.0** | **99.8** | 97.3 | **100.0** | 99.9 | **100.0** | **96.3** |

## C   More Experimental Results

### C.1   More Ablation Results on Three RAVEN Datasets

#### C.1.1   Ablation of DDCP

DDCP is the main reasoning block in this paper. To evaluate its contributions, we replace it with two previously best-performing reasoning blocks such as PredAI [19] and PRB [15]. For a fair comparison, we also equip these three reasoning blocks with an additional CNN gate as in DDCP. The results are summarized in Tab. 11. The results highlight the notable performance gain by DDCP. Compared to PredAI [19] and PRB [15], the average improvement is 1.5% and 1.0% respectively across all configurations over three datasets. This indicates that the dynamic structure in DDCP effectively captures task-specific information, leading to performance improvements.

#### C.1.2   Ablation of MRM

The proposed MRM leverages multi-granular rule mixing to explore diverse rule combinations, thereby enriching feature interactions. To evaluate its effectiveness, we replace MRM with commonly-used structures for feature mixing, *e.g.*, MLP, CNN and Attention. MLP maps features along a single dimension, processing all features uniformly, making it incapable of achieving multi-perspective understanding, while CNN focuses more on local feature extraction, making it challenging to capture global information comprehensively. On the other hand, Attention focuses more on the weight

Table 13: Ablation studies of the three branches in GARB on RAVEN [6], I-RAVEN [8], and RAVEN-FAIR [9] datasets.

| | Components | Accuracy (%) on Different RAVEN Configurations | | | | | | | |
|---|---|---|---|---|---|---|---|---|---|
| | | Avg. | Center | 2×2G | 3×3G | L-R | U-D | O-IC | O-IG |
| O-RVN [6] | **B** | 97.8 | 99.8 | 98.2 | 93.2 | 99.7 | 99.9 | 99.6 | 94.1 |
| | **B+C** | 98.5 | 100.0 | 98.4 | 94.8 | 99.8 | 100.0 | 99.6 | 97.1 |
| | **B+T** | 98.8 | 100.0 | 99.1 | 95.5 | **100.0** | 100.0 | 99.8 | 97.3 |
| | **B+C+T** | **99.1** | **100.0** | **99.3** | **96.2** | 99.9 | **100.0** | **99.9** | **98.2** |
| I-RVN [8] | **B** | 98.1 | 100.0 | 98.6 | 93.7 | 99.9 | 99.8 | 99.7 | 95.2 |
| | **B+C** | 98.6 | 100.0 | 98.5 | 94.8 | 100.0 | 100.0 | 100.0 | 97.1 |
| | **B+T** | 98.9 | 100.0 | 99.2 | 95.6 | 99.9 | **100.0** | 99.8 | 98.1 |
| | **B+C+T** | **99.4** | 100.0 | **99.9** | **97.6** | **100.0** | 99.9 | **100.0** | **98.4** |
| RVN-F [9] | **B** | 98.2 | 100.0 | 98.6 | 96.1 | 99.9 | 100.0 | 99.8 | 93.6 |
| | **B+C** | 98.6 | 100.0 | 99.3 | 96.4 | 99.9 | 100.0 | 100.0 | 95.1 |
| | **B+T** | 98.8 | 100.0 | 99.1 | 97.1 | **100.0** | **100.0** | 99.8 | 95.9 |
| | **B+C+T** | **99.0** | 100.0 | **99.8** | **97.3** | **100.0** | 99.9 | **100.0** | **96.3** |

Table 14: Detailed ablation results of major components on RAVEN [6], I-RAVEN [8], and RAVEN-FAIR [9] datasets.

| | Components | Accuracy (%) on Different RAVEN Configurations | | | | | | | |
|---|---|---|---|---|---|---|---|---|---|
| | | Avg. | Center | 2×2G | 3×3G | L-R | U-D | O-IC | O-IG |
| O-RVN [6] | w/o DDCP | 94.9 | 99.9 | 96.7 | 90.3 | 98.8 | 98.8 | 99.4 | 80.3 |
| | w/o GARB | 95.9 | 100.0 | 98.2 | 91.8 | 99.7 | 99.5 | **100.0** | 82.3 |
| | w/o CNN gate | 97.9 | 100.0 | 98.5 | 93.7 | 99.7 | 99.8 | 98.9 | 94.4 |
| | w/o dynamic gate | 97.8 | 99.8 | 98.2 | 93.2 | 99.7 | 99.9 | 99.6 | 94.1 |
| | w/o MRM | 98.7 | 99.8 | 98.8 | 95.5 | 99.8 | 99.9 | 99.5 | 98.0 |
| | *DSRF-M* | **99.1** | **100.0** | **99.3** | **96.2** | **99.9** | **100.0** | 99.9 | **98.2** |
| I-RVN [8] | w/o DDCP | 95.1 | 100.0 | 97.7 | 91.6 | 99.7 | 99.8 | 99.4 | 77.9 |
| | w/o GARB | 95.6 | 100.0 | 98.6 | 93.0 | 99.9 | 99.8 | 99.6 | 78.9 |
| | w/o CNN gate | 97.8 | 99.8 | 98.8 | 93.0 | 99.9 | 100.0 | 99.7 | 93.5 |
| | w/o dynamic gate | 98.1 | 100.0 | 98.6 | 93.7 | 99.9 | 99.8 | 99.7 | 95.2 |
| | w/o MRM | 98.8 | 100.0 | 98.7 | 95.6 | 100.0 | **100.0** | 99.9 | 97.7 |
| | *DSRF-M* | **99.4** | **100.0** | **99.9** | **97.6** | **100.0** | 99.9 | **100.0** | **98.4** |
| RVN-F [9] | w/o DDCP | 95.6 | 99.5 | 95.0 | 91.3 | 99.3 | 99.6 | 99.3 | 85.3 |
| | w/o GARB | 96.3 | 99.8 | 95.8 | 92.3 | 99.2 | 99.3 | 98.7 | 89.6 |
| | w/o CNN gate | 98.3 | 100.0 | 98.5 | 95.8 | 99.8 | 99.9 | 99.8 | 94.4 |
| | w/o dynamic gate | 98.2 | 100.0 | 98.6 | 96.1 | 99.9 | **100.0** | 99.8 | 93.6 |
| | w/o MRM | 98.6 | 100.0 | 98.2 | 96.5 | 100.0 | 99.9 | 99.7 | 96.3 |
| | *DSRF-M* | **99.0** | **100.0** | **99.8** | **97.3** | **100.0** | 99.9 | **100.0** | **96.3** |

distribution between features rather than the structured decomposition or reorganization of features. The results are shown in Tab. 12.

The proposed MRM showcases its ability to excel in reasoning tasks that involve complex rule compositions, outperforming static or locally focused methods like MLP and CNN, as well as Attention mechanisms that lack explicit feature decomposition. For relatively simpler configurations such as Center, L-R, U-D, and O-IC, all components, including MLP, CNN, and Attention, achieve near-perfect performance, with accuracies close to 100% in many cases. However, for more challenging configurations like 3×3G and O-IG, the proposed MRM consistently outperforms other compared methods. For instance, on the RAVEN dataset [6], the proposed MRM achieves an accuracy of 96.2% and 98.2% on 3×3G and O-IG, respectively, significantly outperforming 95.1% and 96.4% achieved by the Attention mechanism.

Table 15: Detailed ablation results for the proposed DSRF as a plug-and-play reasoning module on three RAVEN datasets [6, 8, 9].

| | Module | | Accuracy (%) on Different RAVEN Configurations | | | | | | | |
|---|---|---|---|---|---|---|---|---|---|---|
| | Perception | Reasoning | Avg. | Center | 2×2G | 3×3G | L-R | U-D | O-IC | O-IG |
| O-RVN [6] | MSE | RM+PM | 84.0 | 98.7 | 72.5 | 52.3 | 99.4 | 99.2 | 99.6 | 66.3 |
| | | DSRF-S | 96.3 | 100.0 | 97.7 | 87.7 | 99.9 | 99.8 | 99.7 | 89.6 |
| | | DSRF-M | 96.4 | 99.9 | 97.7 | 89.6 | 100.0 | 100.0 | 99.5 | 88.5 |
| | HCV | ARR | 87.3 | 99.8 | 71.4 | 65.9 | 99.9 | 99.8 | 98.0 | 76.2 |
| | | DSRF-S | 96.4 | 99.9 | 97.4 | 88.8 | 99.6 | 99.9 | 99.9 | 89.1 |
| | | DSRF-M | 97.5 | 100.0 | 98.8 | 93.1 | 99.9 | 100.0 | 99.9 | 91.0 |
| | RN-4B | PRB | 95.8 | 99.8 | 95.1 | 87.6 | 99.2 | 99.4 | 99.9 | 89.4 |
| | | DSRF-S | 98.6 | 100.0 | 99.3 | 96.1 | 100.0 | 99.9 | 99.9 | 94.9 |
| | | DSRF-M | 99.1 | 100.0 | 99.3 | 96.2 | 99.9 | 100.0 | 99.9 | 98.2 |
| | HPALC | PredAI | 98.8 | 100.0 | 98.8 | 95.3 | 99.9 | 99.8 | 99.9 | 98.0 |
| | | DSRF-S | 99.1 | 100.0 | 99.5 | 96.4 | 99.8 | **100.0** | 100.0 | 98.4 |
| | | DSRF-M | **99.3** | **100.0** | **99.6** | **97.1** | **100.0** | 99.9 | **100.0** | **98.4** |
| I-RVN [8] | MSE | RM+PM | 81.0 | 99.6 | 63.4 | 59.2 | 98.7 | 98.3 | 95.7 | 51.9 |
| | | DSRF-S | 96.7 | 99.8 | 97.1 | 92.0 | 99.6 | 99.6 | 99.6 | 89.2 |
| | | DSRF-M | 96.9 | 100.0 | 97.7 | 90.6 | 99.8 | 99.8 | 99.8 | 90.9 |
| | HCV | ARR | 93.9 | 99.9 | 96.2 | 75.5 | 99.4 | 99.6 | 99.5 | 87.3 |
| | | DSRF-S | 96.5 | 100.0 | 99.2 | 91.5 | 100.0 | 99.9 | 99.8 | 85.2 |
| | | DSRF-M | 97.2 | 100.0 | 99.2 | 94.9 | 99.9 | 99.9 | 99.9 | 86.7 |
| | RN-4B | PRB | 96.5 | 99.9 | 97.8 | 91.2 | 99.7 | 99.7 | 99.6 | 87.7 |
| | | DSRF-S | 98.8 | 100.0 | 99.5 | 96.7 | 100.0 | 99.9 | 99.8 | 95.8 |
| | | DSRF-M | 99.4 | 100.0 | 99.9 | 97.6 | 100.0 | 99.9 | 100.0 | 98.4 |
| | HPALC | PredAI | 99.4 | 100.0 | 99.9 | 97.4 | 99.9 | 100.0 | 100.0 | **98.8** |
| | | DSRF-S | 99.4 | 100.0 | 99.5 | 98.2 | 100.0 | 100.0 | 100.0 | 98.5 |
| | | DSRF-M | **99.5** | **100.0** | **99.9** | **98.4** | **100.0** | **100.0** | **100.0** | 98.5 |
| RVN-F [9] | MSE | RM+PM | 86.8 | 97.0 | 72.7 | 69.5 | 98.7 | 98.9 | 97.6 | 73.3 |
| | | DSRF-S | 96.5 | 100.0 | 93.6 | 89.7 | 99.1 | 100.0 | 99.9 | 93.4 |
| | | DSRF-M | 97.9 | 99.9 | 98.8 | 92.7 | 100.0 | 100.0 | 99.7 | 94.1 |
| | HCV | ARR | 95.4 | 99.8 | 92.9 | 87.9 | 99.8 | 99.6 | 99.7 | 88.5 |
| | | DSRF-S | 97.9 | 99.8 | 98.9 | 93.0 | 100.0 | 99.8 | 99.6 | 94.3 |
| | | DSRF-M | 98.1 | 99.9 | 99.0 | 94.0 | 99.8 | 99.9 | 99.7 | 94.8 |
| | RN-4B | PRB | 97.1 | 99.8 | 97.3 | 92.6 | 99.7 | 99.5 | 99.7 | 91.2 |
| | | DSRF-S | 98.8 | 100.0 | 99.6 | 96.2 | 100.0 | 99.9 | 100.0 | 96.2 |
| | | DSRF-M | 99.0 | 100.0 | 99.8 | 97.3 | 100.0 | 99.9 | **100.0** | 96.3 |
| | HPALC | PredAI | 98.6 | 100.0 | 99.4 | 96.9 | 99.9 | 99.9 | 99.7 | 94.2 |
| | | DSRF-S | 99.2 | 99.9 | 99.6 | 97.5 | **100.0** | 99.9 | 99.9 | 97.8 |
| | | DSRF-M | **99.4** | **100.0** | **99.9** | **98.3** | 99.9 | **100.0** | 99.9 | **97.8** |

### C.1.3   Ablation of Three Branches in GARB

In Tab. 13, we conduct an ablation study to evaluate the contributions of the three branches in the proposed GARB module: **B** represents the **base** branch for fundamental feature extraction, **C** denotes the **context**-enhancement branch that emphasizes context features, and **T** stands for the **target**-enhancement branch designed to refine target-specific features. The results are summarized in Tab. 13.

The results demonstrate a clear progressive performance improvement as more branches are integrated. The base branch alone achieves an average accuracy of 98.0% over three datasets across different configurations. Adding the context-enhancement branch (**B+C**) improves the average accuracy

Table 16: Comparison with state-of-the-art models on the original RAVEN dataset [6], ‡ denotes that the original method is based on contrasting over candidate answers.

| Models | Acc.(%) on RAVEN Dataset [6] Under In-Distribution Settings | | | | | | | |
|---|---|---|---|---|---|---|---|---|
| | Avg. | Center | 2×2G | 3×3G | L-R | U-D | O-IC | O-IG |
| ‡ HCV-ARR (AAAI'23) [13] | 96.0 | 99.4 | 86.9 | 89.1 | 99.9 | 99.9 | 99.8 | 96.8 |
| ‡ MRNet (CVPR'21) [9] | 96.6 | 99.9 | 97.8 | 91.2 | 99.7 | 99.7 | 99.6 | 87.7 |
| MRNet (CVPR'21) [9] | 84.0 | 98.7 | 72.5 | 52.3 | 99.4 | 99.2 | 99.6 | 66.3 |
| HCV-ARR (AAAI'23) [14] | 87.3 | 99.8 | 71.4 | 65.9 | 99.9 | 99.8 | 98.0 | 76.2 |
| SCAR (AAAI'24) [17] | 92.8 | 98.7 | 80.4 | 92.9 | 99.1 | 99.3 | 98.2 | 81.2 |
| AlgeMR (CVPR'23) [13] | 92.9 | 98.8 | 91.9 | 93.1 | 99.2 | 99.1 | 98.2 | 70.1 |
| PredRNet (ICML'23) [15] | 95.8 | 99.8 | 95.1 | 87.6 | 99.2 | 99.4 | 99.9 | 89.4 |
| HP$^2$AI (ACM MM'24) [19] | 98.8 | 100.0 | 98.8 | 95.3 | 99.9 | 99.8 | 99.9 | 98.0 |
| *DSRF-S* (Ours) | 98.6 | 100.0 | 99.3 | 96.1 | **100.0** | 99.9 | 99.9 | 94.9 |
| *DSRF-M* (Ours) | 99.1 | 100.0 | 99.3 | 96.2 | 99.9 | 100.0 | 99.9 | 98.2 |
| *DSRF-L* (Ours) | **99.1** | **100.0** | **99.3** | **96.5** | 99.9 | **100.0** | **99.9** | **98.3** |

Table 17: Comparison with state-of-the-art models on I-RAVEN [8] and RAVEN-FAIR [9] datasets.

| Models | Acc.(%) on I-RAVEN [8]/RAVEN-FAIR [9] Datasets Under In-Distribution Settings | | | | | | | |
|---|---|---|---|---|---|---|---|---|
| | Avg. | Center | 2×2G | 3×3G | L-R | U-D | O-IC | O-IG |
| MRNet (CVPR'21) [9] | 81.0/86.8 | 99.6/97.0 | 63.4/72.7 | 59.2/69.5 | 98.7/98.7 | 98.3/98.9 | 95.7/97.6 | 51.9/73.3 |
| AlgeMR (CVPR'23) [13] | 93.2/94.3 | 99.5/99.8 | 89.6/93.2 | 89.7/88.0 | 99.7/99.8 | 99.5/99.8 | 99.6/99.9 | 74.7/79.6 |
| HCV-ARR (AAAI'23) [14] | 93.9/95.4 | 99.9/99.8 | 96.2/92.9 | 75.5/87.9 | 99.4/99.8 | 99.6/99.6 | 99.5/99.7 | 87.3/88.5 |
| SCAR (AAAI'24) [17] | 94.7/93.9 | 99.1/98.6 | 95.7/93.1 | 80.4/81.3 | 99.3/99.8 | 99.3/99.8 | 98.2/99.3 | 91.2/85.7 |
| PredRNet (ICML'23) [15] | 96.5/97.1 | 99.9/99.8 | 97.8/97.3 | 91.2/92.6 | 99.7/99.7 | 99.7/99.5 | 99.6/99.7 | 87.7/91.2 |
| HP$^2$AI (ACM MM'24) [19] | 99.4/98.6 | 100.0/100.0 | 99.9/99.4 | 97.4/96.9 | 99.9/99.9 | 100.0/99.9 | 100.0/99.7 | 98.8/94.2 |
| *DSRF-S* (Ours) | 98.8/98.8 | 100.0/100.0 | 99.5/99.6 | 96.7/96.2 | 100.0/100.0 | 99.9/99.9 | 99.8/100.0 | 95.8/96.2 |
| *DSRF-M* (Ours) | 99.4/99.0 | 100.0/100.0 | 99.9/99.8 | 97.6/97.3 | **100.0**/100.0 | 99.9/99.9 | **100.0**/100.0 | 98.4/96.3 |
| *DSRF-L* (Ours) | **99.4/99.1** | **100.0/100.0** | **99.9/99.8** | **97.6/97.4** | 99.9/**100.0** | **100.0/100.0** | 99.9/**100.0** | **98.8/96.5** |

to 98.6%, highlighting the benefit of incorporating contextual features. Incorporating the target-enhancement branch (**B+T**) provides a more significant boost, achieving an average accuracy of 98.8%, indicating the importance of target-specific adjustments. Finally, the full three-branch structure (**B+C+T**) achieves the highest average accuracy of 99.2%, demonstrating the complementary strengths of the three branches and their effectiveness in capturing task-specific information for enhanced reasoning performance.

### C.1.4 Detailed Ablation of Major Components

In Tab. 5 of the manuscript, we summarize the ablation study of the major components on the three RAVEN datasets [6, 8, 9]. Now, we present the detailed ablation study results in Tab. 14, showcasing the contributions of key components, including DDCP, GARB, CNN gate, dynamic gate, and MRM. As shown in Tab. 14, we observe that for simpler configurations such as Center, L-R, and U-D on all three datasets, our method achieves an accuracy of 100% in most cases even when some of our components are removed. However, for more challenging configurations such as 2×2G, 3×3G, and O-IG, removing specific components leads to significant performance drops, particularly those related to dynamic reasoning. For example, for O-IG on the RAVEN dataset [6], removing the DDCP module results in a sharp decline from 98.2% to 80.3%, highlighting the importance of DDCP in dynamically identifying and contrasting features. Similarly, for 3×3G on the RAVEN-FAIR dataset [9], removing the dynamic gate reduces accuracy from 97.3% to 96.1%, demonstrating the critical role of dynamic adaptability for handling complex rules.

### C.1.5 Detailed Results of DSRF as Plug-and-Play Reasoning Module

Due to the page limit, we only summarize the average performance for DSRF serving as a plug-and-play reasoning module across different configurations in Tab. 6 of the manuscript. Here, we

Table 18: Comparison with state-of-the-art models under OOD settings on three RAVEN datasets [6, 8, 9].

| Models | Avg. | Center | 2×2G | 3×3G | L-R | U-D | O-IC | O-IG |
|---|---|---|---|---|---|---|---|---|
| **Training on RAVEN [6] while testing on I-RAVEN [8] / RAVEN-FAIR [9] Datasets** | | | | | | | | |
| MRNet (CVPR'21) [9] | 65.8/81.2 | 86.8/96.1 | 40.3/58.9 | 35.3/54.3 | 87.2/99.3 | 86.1/98.9 | 88.4/97.6 | 36.7/63.3 |
| HCV-ARR (AAAI'23) [14] | 79.4/91.9 | 99.4/99.9 | 54.6/84.1 | 51.2/80.4 | 98.9/99.7 | 98.7/99.4 | 96.3/99.5 | 60.4/80.2 |
| SCAR (AAAI'24) [17] | 84.7/95.9 | 99.8/99.9 | 65.5/95.1 | 59.4/86.6 | 99.7/99.8 | 99.8/99.8 | 99.9/99.6 | 68.6/90.3 |
| HP$^2$AI (ACM MM'24) [19] | 87.7/96.9 | 100.0/100.0 | 72.1/93.7 | 63.1/91.4 | 100.0/99.9 | 100.0/99.9 | 99.9/99.9 | 78.2/93.2 |
| PredRNet (ICML'23) [15] | 89.2/97.9 | 100.0/100.0 | 75.4/95.4 | 68.8/94.9 | 100.0/99.9 | 99.9/99.9 | 100.0/99.9 | 80.9/95.1 |
| **DSRF-S** (Ours) | 95.1/98.5 | 100.0/100.0 | 91.3/99.5 | 80.9/95.2 | 100.0/100.0 | 100.0/99.9 | 100.0/100.0 | 93.7/95.1 |
| **DSRF-M** (Ours) | 95.6/98.7 | 100.0/100.0 | 91.9/99.7 | 82.5/96.2 | 100.0/100.0 | 100.0/100.0 | **100.0**/100.0 | 94.5/95.2 |
| **DSRF-L** (Ours) | **95.9**/**98.9** | 100.0/100.0 | **92.5**/**99.9** | **84.0**/**96.9** | 100.0/100.0 | 100.0/100.0 | 99.9/**100.0** | **95.2**/**95.8** |
| **Training on I-RAVEN [8] while testing on RAVEN [6] / RAVEN-FAIR [9] Datasets** | | | | | | | | |
| MRNet (CVPR'21) [9] | 76.5/78.6 | 98.7/98.5 | 60.7/58.5 | 47.6/48.2 | 95.7/96.3 | 96.3/97.4 | 91.9/96.9 | 44.3/54.2 |
| HCV-ARR (AAAI'23) [14] | 81.4/89.4 | 99.7/98.9 | 61.7/79.9 | 49.5/69.8 | 98.8/98.3 | 98.9/98.9 | 98.1/98.5 | 63.3/81.3 |
| HP$^2$AI (ACM MM'24) [19] | 89.9/94.0 | 99.2/99.5 | 77.2/86.6 | 74.1/83.0 | 99.9/100.0 | 99.7/100.0 | 99.8/99.9 | 79.5/89.4 |
| SCAR (AAAI'24) [17] | 91.9/94.5 | 99.1/99.9 | 84.8/87.8 | 82.4/83.9 | 99.6/99.9 | 99.5/99.8 | 99.6/99.9 | 78.6/90.1 |
| PredRNet (ICML'23) [15] | 93.9/97.3 | 99.3/99.4 | 93.9/97.3 | 84.1/93.0 | 99.6/99.9 | 99.5/99.7 | 99.4/100.0 | 81.5/92.0 |
| **DSRF-S** (Ours) | 95.9/98.2 | 100.0/100.0 | 96.3/98.2 | 87.6/94.7 | 99.8/100.0 | 100.0/100.0 | 99.6/99.9 | 87.9/95.0 |
| **DSRF-M** (Ours) | 97.4/98.7 | 100.0/100.0 | 96.8/97.5 | 92.3/95.7 | 99.9/100.0 | 100.0/**100.0** | 99.6/99.9 | 93.4/**98.0** |
| **DSRF-L** (Ours) | **97.9**/**98.9** | 100.0/100.0 | **97.6**/**98.5** | **93.7**/**96.6** | 100.0/100.0 | 100.0/99.9 | **99.9**/100.0 | 94.3/97.4 |
| **Training on RAVEN-FAIR [9] while testing on RAVEN [6] / I-RAVEN [8] Datasets** | | | | | | | | |
| MRNet (CVPR'21) [9] | 78.8/73.4 | 93.9/91.1 | 61.3/53.5 | 49.2/41.3 | 93.1/89.1 | 92.1/89.4 | 95.9/89.1 | 66.1/60.5 |
| SCAR (AAAI'24) [17] | 91.3/85.6 | 98.4/98.3 | 84.3/64.3 | 73.6/51.5 | 99.4/99.4 | 99.4/99.3 | 99.0/99.4 | 84.8/87.3 |
| HCV-ARR (AAAI'23) [14] | 94.1/87.3 | 99.1/98.8 | 88.5/68.5 | 79.4/56.2 | 99.7/99.4 | 99.5/99.5 | 99.2/99.5 | 93.1/89.1 |
| HP$^2$AI (ACM MM'24) [19] | 95.3/90.0 | 99.2/99.7 | 90.3/72.9 | 84.9/65.7 | 99.8/99.8 | 99.5/99.7 | 99.3/99.9 | **94.1**/**92.8** |
| PredRNet (ICML'23) [15] | 96.2/91.4 | 99.6/100.0 | 97.4/86.5 | 88.9/68.3 | 99.6/100.0 | 99.1/99.4 | 99.6/99.8 | 89.7/86.3 |
| **DSRF-S** (Ours) | 97.3/95.4 | 99.9/99.9 | 98.1/96.0 | 90.8/82.3 | 99.6/100.0 | 99.6/100.0 | 99.6/99.9 | 93.8/90.4 |
| **DSRF-M** (Ours) | 97.8/95.8 | 100.0/100.0 | 97.9/96.9 | 93.3/82.7 | 99.9/**100.0** | 99.9/**100.0** | 99.8/**100.0** | 93.8/90.7 |
| **DSRF-L** (Ours) | **98.1**/**96.5** | 100.0/100.0 | **98.1**/**97.7** | **94.9**/**88.9** | 99.9/99.9 | 100.0/99.9 | **99.9**/99.9 | 93.7/89.5 |

supplement the detailed evaluation results on all configurations in Tab. 15. For most configurations, the proposed method demonstrates significant performance gains when combined with different visual perception modules. Specifically, for easier configurations such as Center, L-R, U-D, and O-IC, our reasoning module, when combined with certain perception modules, achieves perfect reasoning with 100% accuracy. For more challenging configurations like 3×3G, O-IG, and 2×2G, our reasoning module also delivers significant improvements. For example, on the 3×3G configuration, DSRF-M achieves 97.1% accuracy with HPALC [19] on the RAVEN dataset [6], a substantial enhancement compared to the 52.3% achieved by MSE with RM+PM [9].

## C.2   Detailed Results on Three RAVEN Datasets Under In-Distribution Settings

In Tab. 1 of the manuscript, we summarize the comparison results on the three RAVENs over all configurations. Here, we supplement the results for all configurations on RAVEN [6], I-RAVEN [8], and RAVEN-FAIR [9] datasets. Detailed results on three RAVEN datasets are presented in Tab. 16 and Tab. 17. From these two tables, we can observe the following. 1) When the shortcut in RAVEN [6] is not leveraged, MRNet [9] and HCV-ARR [14] both experience a notable performance drop. In contrast, the proposed DSRF-L achieves the highest average accuracy of 99.1% without relying on the shortcut, demonstrating the superiority of its reasoning capability. 2) On simple configurations, such as Center, 2×2G, L-R, U-D, and O-IC, our three methods all comprehensively understand the underlying rules, achieving accuracies exceeding 99.0% across three RAVEN datasets [6, 8, 9]. 3) However, on challenging configurations such as 3×3G, all compared methods perform relatively poorly. In contrast, the proposed DSRF-L obtains the highest average accuracies of 96.5%, 97.6%, and 97.4% on RAVEN [6], I-RAVEN [8] and RAVEN-FAIR [9], respectively, improving the second-best model HP$^2$AI [19] by 1.2%, 0.2% and 0.5%. These results demonstrate the robustness and effectiveness of the model's dynamic reasoning ability. 4) On the challenging O-IG configuration, the proposed DSRF-L obtains the largest performance gain over the second-best model HP$^2$AI [19] by 2.3% on the RAVEN-FAIR dataset [9]. This highlights the reasoning ability of the DDCP to dynamically encapsulate complex relational patterns.

## C.3 Detailed Results on Three RAVEN Datasets Under Out-Of-Distribution Settings

We report the comparison results on the three RAVEN datasets [6, 8, 9] under OOD settings in Tab. 2 of the manuscript. Here, we supplement the results on different configurations to show the model's generalization ability to transfer knowledge across different settings, where the model is trained on one dataset while tested on the other two. Detailed results under OOD settings are presented in Tab. 18. From this table, we can observe the following. 1) Our three models consistently and significantly outperform all existing models across almost all configurations under all OOD settings. 2) The proposed DSRF-L exhibits consistent performance across 7 configurations, demonstrating accuracy fluctuations of 16.0% and 4.2% under RAVEN OOD setting, 6.3% and 3.4% under I-RAVEN OOD setting, and 6.3% and 11.1% under RAVEN-FAIR OOD setting. Specifically, considering RAVEN OOD setting where the accuracy fluctuations are the highest, our DSRF-L still outperforms the second-best model PredRNet [15] with 15.2% and 0.9% reduction in accuracy fluctuation, illustrating the robustness of our models across diverse testing configurations.

## D Failure Case Analysis

Fig. 10 presents failure cases to visually illustrate the benefit of our method, in contrast to PredR-Net's [15]. In Case 1, PredRNet [15] overfits the RAVEN dataset [6] by selecting the most common option from the answer panels, where eight objects have varying colors, failing to derive the underlying abstract rules, whereas DSRF-M effectively identifies the correct answer by capturing the rule-based patterns dynamically. In Case 2, the task involves complex rule combinations, making it challenging for PredRNet's [15] static "predict-and-verify" approach to reason about the correct solution. In contrast, DSRF-M leverages dynamic attention mechanisms to focus on the layout structure, enabling it to reason and select the correct answer. Case 3 also represents a more complex problem due to the combination of multiple rules, which results in a significantly challenging task. The proposed DSRF could solve it but PredRNet [15] cannot.

In Cases 4, 5, and 6, both methods fail to select the correct answer. In particular, Case 4 and Case 5 both involve highly complex layouts, where it is challenging to precisely derive the intricate arrangement of objects. In Case 6, there are two irrelevant attributes `Position` and `Type` that may confuse both models. Indeed, reasoning over complex layout and being robust to irrelevant attributes are remaining challenges, which are worth to further explore in future.

## E Limitation

Although the Multi-View Reasoning Pyramid (MVRP) enables scalable and structured reasoning by mapping features into multiple views, the current implementation adopts a manually predefined number of views per layer (*e.g.*, "2-1", "4-2-1", "8-4-1"). While these settings have shown strong performance across benchmarks, they may not be optimal for all tasks. Automatically discovering the number and composition of views could further improve the model's adaptability. In particular, different problem types may benefit from different view granularities, and a one-size-fits-all design might limit the model's flexibility in some scenarios. Additionally, for certain complex tasks, fine-grained or hierarchical view configurations could be beneficial but have not yet been explored in the current framework. Investigating how view configurations interact with rule complexity would provide valuable insights for future enhancements.

## F Summary of Network Architecture

We provide the detailed architecture of our proposed DDCP in Tab. 19, MRM in Tab. 20 and core component GARB in Tab. 21. For readability, we do not include batch normalization, layer normalization, dropout layer and activation functions in the structure. For example, F(8, 32, 3, 25) represents a feature with a group size of 8, 32 channels, 3 context features in a row, and a token size of 25. The group size is 8 because we concatenate 8 answer images with the question panel. Additionally, before transforming into multiple heads in GARB, our features are three-dimensional, such as F(8, 25, 32), which represents a batch size of 8, a token size of 25, and 32 channels. More details will be provided in our released code upon the paper acceptance.

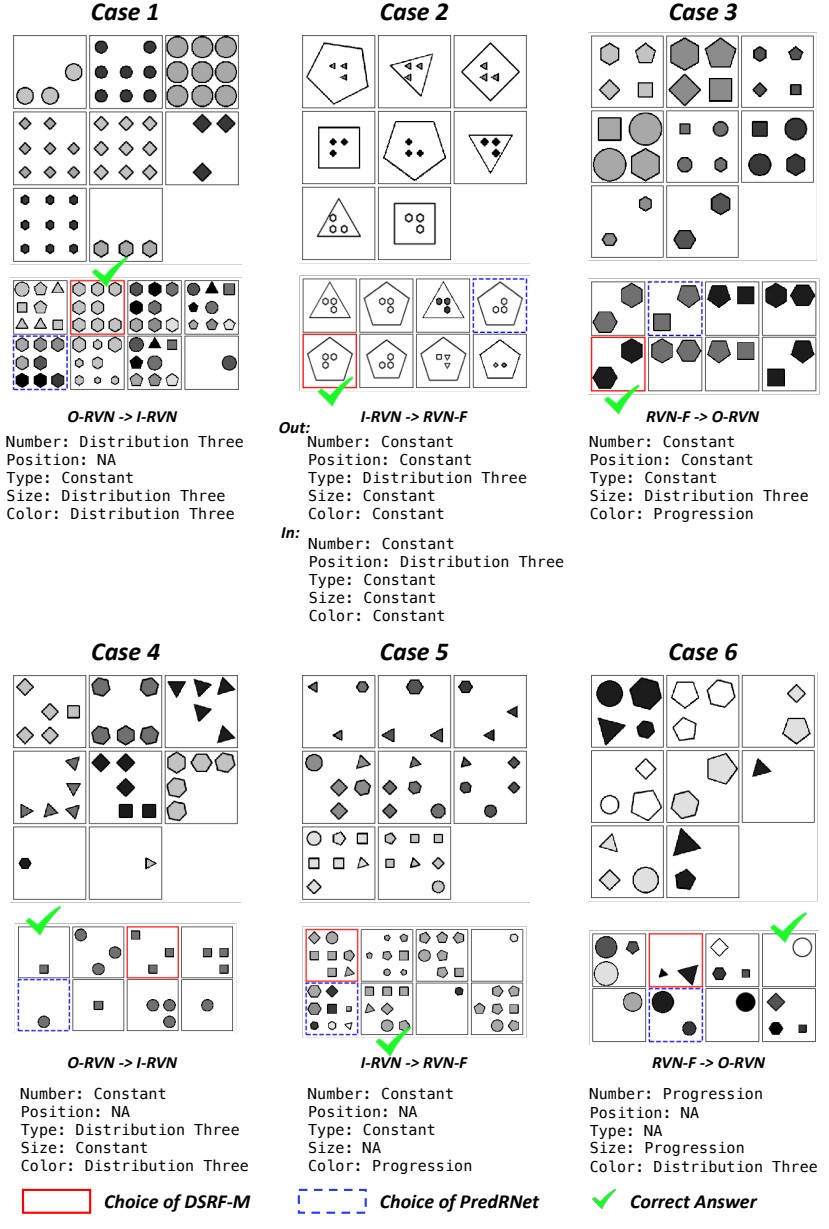

Figure 10: Failure cases on RAVEN datasets under OOD settings. The choices of DSRF-M and PredRNet [15] are compared.

Table 19: Detailed network architecture of the proposed DDCP, with the number of parameter channels $\alpha$, the kernel size $\beta$. To better illustrate the CNN gate, we denote the G as the gate matrix.

| Function | Operations | Input | Output |
|---|---|---|---|
| | Permute | F(8, 32, 9, 25) | F(8, 9, 25, 32) |
| | Linear($\alpha$=64) | F(8, 9, 25, 32) | F(8, 9, 25, 64) |
| | Split | F(8, 9, 25, 64) | F(8, 9, 25, 32) &G(8, 9, 25, 32) |
| | Permute | F(8, 9, 25, 32) | F(8, 32, 9, 25) |
| | Split | F(8, 32, 9, 25) | F(8, 32, 3, 25)×3 |
| Reason 1st Row | GARB($\alpha$=32) | F(8, 32, 3, 25) | F(8, 32, 3, 25) |
| Reason 2nd Row | GARB($\alpha$=32) | F(8, 32, 3, 25) | F(8, 32, 3, 25) |
| Reason 3rd Row | GARB($\alpha$=32) | F(8, 32, 3, 25) | F(8, 32, 3, 25) |
| | Concat | F(8, 32, 3, 25)×3 | F(8, 32, 9, 25) |
| | Permute | F(8, 32, 9, 25) | F(8, 9, 25, 32) |
| **CNN Gate** | | | |
| | Conv2d($\alpha$=36, $\beta$=(3, 3)) | G(8, 9, 25, 32) | G(8, 36, 25, 32) |
| | Conv2d($\alpha$=9, $\beta$=(3, 3)) | G(8, 36, 25, 32) | G(8, 9, 25, 32) |
| | Linear($\alpha$=32) | G(8, 9, 25, 32) | G(8, 9, 25, 32) |
| | Hadamard Product | F(8, 9, 25, 32)&G(8, 9, 25, 32) | F(8, 9, 25, 32) |
| | Permute | F(8, 9, 25, 32) | F(8, 32, 9, 25) |
| | Conv2d($\alpha$=128, $\beta$=(3, 3)) | F(8, 32, 9, 25) | F(8, 128, 9, 25) |
| | Conv2d($\alpha$=32, $\beta$=(3, 3)) | F(8, 128, 9, 25) | F(8, 32, 9, 25) |
| Residual | Add | F(8, 32, 9, 25)×2 | F(8, 32, 9, 25) |

Table 20: Detailed network architecture of the proposed MRM. $\alpha$ denotes the number of channels. To illustrate the parallel structure in the model, we denote the feature for the channel-wise linear map as C and the feature for the token-wise linear map as T.

| Function | Operations | Input | Output |
|---|---|---|---|
| Token-wise Map | Linear($\alpha$=25) | F(8, 32, 9, 25) | T(8, 32, 9, 25) |
| | Permute | T(8, 32, 9, 25) | T(8, 9, 25, 32) |
| | Permute | F(8, 32, 9, 25) | F(8, 9, 25, 32) |
| Channel-wise Map | Linear($\alpha$=32) | F(8, 9, 25, 32) | C(8, 9, 25, 32) |
| | Add | T(8, 9, 25, 32)&C(8, 9, 25, 32) | F(8, 9, 25, 32) |
| | Reshape&Permute | F(8, 9, 25, 32) | F(8, 25, 32, 3, 3) |
| Row-wise Map | Linear($\alpha$=3) | F(8, 25, 32, 3, 3) | F(8, 25, 32, 3, 3) |
| | Reshape&Permute | F(8, 25, 32, 3, 3) | F(8, 25, 32, 9) |
| Sample-wise Map | Linear($\alpha$=9) | F(8, 25, 32, 9) | F(8, 25, 32, 9) |
| | Permute | F(8, 25, 32, 9) | F(8, 32, 9, 25) |
| | Conv2d($\alpha$=128, $\beta$=(3, 3)) | F(8, 32, 9, 25) | F(8, 128, 9, 25) |
| | Conv2d($\alpha$=32, $\beta$=(3, 3)) | F(8, 128, 9, 25) | F(8, 32, 9, 25) |
| Residual | Add | F(8, 32, 9, 25)×2 | F(8, 32, 9, 25) |

Table 21: Detailed network architecture of the proposed GARB, with the number of parameter channels $\alpha$, the kernel size $\beta$. W is the learnable matrix. GQ and GK denote the two Gram matrices. DGQ and DGK are Q-wise and K-wise dynamic gates. DQ and DK represent two enhanced features after dynamic gates. D is the domain difference. E is the prediction error.

| Function | Operations | Input | Output |
|---|---|---|---|
| | Split | F(8, 32, 3, 25) | F(8, 32, 2, 25) &F(8, 32, 1, 25) |
| Downsample | Conv2d($\alpha$=32, $\beta$=(2, 1)) | F(8, 32, 2, 25) | F(8, 32, 1, 25) |
| Q Generation | Reshape&Permute | F(8, 32, 1, 25) | F(8, 25, 32) |
| K Generation | Reshape&Permute | F(8, 32, 1, 25) | F(8, 25, 32) |
| V Generation | Reshape&Permute | F(8, 32, 1, 25) | F(8, 25, 32) |
| Q Mapping | Linear($\alpha$=32) | F(8, 25, 32) | Q(8, 25, 32) |
| K Mapping | Linear($\alpha$=32) | F(8, 25, 32) | K(8, 25, 32) |
| V Mapping | Linear($\alpha$=32) | F(8, 25, 32) | V(8, 25, 32) |
| Q Multi-head | Reshape&Permute | Q(8, 25, 32) | Q(8, 8, 25, 4) |
| K Multi-head | Reshape&Permute | K(8, 25, 32) | K(8, 8, 25, 4) |
| V Multi-head | Reshape&Permute | V(8, 25, 32) | V(8, 8, 25, 4) |
| **Basic Branch** | | | |
| Gram GQ | Dot Product | Q(8, 8, 25, 4)&Q(8, 8, 25, 4) | GQ(8, 8, 25, 25) |
| Gram GK | Dot Product | K(8, 8, 25, 4)&K(8, 8, 25, 4) | GK(8, 8, 25, 25) |
| GQ Highlight | Hadamard Product | GQ(8, 8, 25, 25)&W(8, 8, 25, 25) | GQ(8, 8, 25, 25) |
| GK Highlight | Hadamard Product | GK(8, 8, 25, 25)&W(8, 8, 25, 25) | GK(8, 8, 25, 25) |
| Difference D | Minus | GQ(8, 8, 25, 25)&GK(8, 8, 25, 25) | D(8, 8, 25, 25) |
| | Conv2d($\alpha$=32,$\beta$=(3,3)) | D(8, 8, 25, 25) | D(8, 32, 25, 25) |
| | Conv2d($\alpha$=8,$\beta$=(3,3)) | D(8, 32, 25, 25) | D(8, 8, 25, 25) |
| **Context-enhanced Branch** | | | |
| DGK Dynamic Gate | Dot Product | K(8, 8, 25, 4)&W(8, 8, 25, 4) | DGK(8, 8, 25, 25) |
| DK Highlight | Hadamard Product | D(8, 8, 25, 25)&DGK(8, 8, 25, 25) | DK(8, 8, 25, 25) |
| | Conv2d($\alpha$=32,$\beta$=(3,3)) | DK(8, 8, 25, 25) | DK(8, 32, 25, 25) |
| | Conv2d($\alpha$=8,$\beta$=(3,3)) | DK(8, 32, 25, 25) | DK(8, 8, 25, 25) |
| **Target-enhanced Branch** | | | |
| DGQ Dynamic Gate | Dot Product | Q(8, 8, 25, 4)&W(8, 8, 25, 4) | DGQ(8, 8, 25, 25) |
| DQ Highlight | Hadamard Product | D(8, 8, 25, 25)&DGQ(8, 8, 25, 25) | DQ(8, 8, 25, 25) |
| | Conv2d($\alpha$=32,$\beta$=(3,3)) | DQ(8, 8, 25, 25) | DQ(8, 32, 25, 25) |
| | Conv2d($\alpha$=8,$\beta$=(3,3)) | DQ(8, 32, 25, 25) | DQ(8, 8, 25, 25) |
| Prediction { | Plus | D&DK&DQ(8, 8, 25, 25)×3 | D(8, 8, 25, 25) |
| | Dot Product | D(8, 8, 25, 25)&V(8, 8, 25, 4) | F(8, 8, 25, 4) |
| | Reshape&Permute | F(8, 8, 25, 4) | F(8, 25, 32) |
| | Linear($\alpha$=32) | F(8, 25, 32) | F(8, 25, 32) |
| | Reshape&Permute | F(8, 25, 32) | F(8, 32, 1, 25) |
| Error E | Minus | F(8, 32, 1, 25)&F(8, 32, 1, 25) | E(8, 32, 1, 25) |
| | Concat | F(8, 32, 2, 25)&E(8, 32, 1, 25) | F(8, 32, 3, 25) |

