# OpenReview forum: "DSRF: A Dynamic and Scalable Reasoning Framework for Solving RPMs"
_NeurIPS.cc/2025/Conference — NeurIPS 2025 poster_

### Official Review · Reviewer_Yeks · 2025-06-18

**Clarity:** 2
**Significance:** 2
**Originality:** 2
**Rating:** 4
**Confidence:** 3

**Summary:**

The paper presents a framework called the Dynamic and Scalable Reasoning Framework (DSRF) aimed at solving Raven’s Progressive Matrices (RPMs), a benchmark for abstract visual reasoning (AVR). The core contributions of the paper lie in two main components: the Multi-View Reasoning Pyramid (MVRP), which enables the model to scale by expanding the network across multiple views to capture complex attribute combinations, and the Dynamic Domain-Contrast Prediction (DDCP) block, which utilizes a gating mechanism and Gram matrices to dynamically handle domain shifts and task-specific relationships. The authors argue that DSRF addresses key challenges in AVR, particularly scalability and adaptability, by widening the network rather than deepening it. Extensive experiments on six AVR tasks demonstrate DSRF’s superior performance over state-of-the-art models in various settings.

**Questions:**

1.The author claims that each view in the MVRP focuses on distinct attribute combinations. How is this enforced or encouraged during training? Is there any orthogonality loss, entropy regularization, or clustering analysis to show that views specialize on different features?

2.The scalability of DSRF is presented as a strength, yet the performance improvement from DSRF-M to DSRF-L is marginal given the increase in computational cost. Can the authors provide more analysis or visualization to justify whether the scalability is practically meaningful?

3.The argument that deeper reasoning networks tend to overfit is used as a central motivation for the proposed framework. However, no empirical evidence is provided to support this claim. For example, what happens to performance when increasing the depth of a PredRNet or HP2AI baseline beyond 4 blocks?

4.Including results from representative LLMs on AVR would strengthen the empirical evaluation and provide a clearer comparison.

5.The paper introduces MVRP, DDCP, and GARB as hierarchically organized components, yet presents them under the same heading level. Could the authors clarify their structural relationships more explicitly, perhaps by adjusting the heading hierarchy to better reflect their conceptual roles?

**Ethical Concerns:**

["NO or VERY MINOR ethics concerns only"]

**Final Justification:**

After reviewing the authors' rebuttal, I find that they have addressed several of my key concerns, particularly regarding the scalability analysis, specialization of views in MVRP, and the empirical evidence for their claims about overfitting in deeper reasoning networks. The additional clarifications and supplementary results, including comparisons to representative LLMs, have strengthened the empirical evaluation and reinforced the practical relevance of the proposed framework. While some aspects of novelty remain incremental, the methodological integration is well-motivated and demonstrates consistent and competitive performance across diverse AVR tasks. Given these clarifications and the solid experimental results, I now view this as a technically sound and valuable contribution to the AVR community, and I recommend acceptance.

**Limitations:**

Yes.

**Paper Formatting Concerns:**

No formatting issues noticed. The paper follows NeurIPS 2025 guidelines.

**Quality:**

3

**Strengths And Weaknesses:**

Strengths:

1.Competitive Performance: The proposed DSRF consistently outperforms prior methods, including recent state-of-the-art models, with noticeable margins, especially in challenging OOD settings.

2.Modularity and Transferability: The fact that DSRF can be plugged into different perception backbones and still improve performance shows the potential generality and applicability of the method.

Weaknesses:

1.Limited Novelty: Much of the underlying technology, such as the pyramid structure and gated attention mechanisms, relies on well-established deep learning techniques. While the combination of these methods is valid, the novelty of the contributions remains relatively limited. The paper does not provide sufficient groundbreaking innovations to distinguish it from other recent works in AVR.

2.Insufficient analysis of scalability: While the scalability claim is central to the paper, the marginal performance gain from DSRF-M to DSRF-L is relatively small (e.g., 97.3% to 97.7% in Table 2), despite a considerable increase in FLOPs and parameters. The paper lacks a deeper justification of this trade-off and whether the scalability is truly meaningful in practice.

3.Unsubstantiated Overfitting Claim: The authors argue that stacking deep reasoning modules in existing models often leads to overfitting. However, no experimental evidence or quantitative comparison is provided to support this claim.

4.Lack of Comparison with LLMs: Although the authors state that AVR has become a standard benchmark for evaluating the reasoning capabilities of large language models (LLMs), the experimental section does not include any comparison with such models. This omission weakens the paper’s claim regarding the effectiveness and competitiveness of DSRF in the broader context of reasoning benchmarks.

5.Inconsistent Structural Presentation: The proposed framework introduces MVRP, DDCP, and GARB as distinct module levels, yet presents them under the same heading tier in the paper’s organization. This parallel presentation of conceptually hierarchical components may lead to confusion about their structural relationships and respective roles within the architecture.

---

> ### Author Rebuttal · Authors · 2025-07-31
>
> **W1: While combination is valid, novelty remains relatively limited.**
>
> While our framework builds upon certain established deep learning components, its core design is fundamentally different from prior architectures.
>
> - **MVRP differs significantly from spatial pyramids.**
>
> MVRP diverges from spatial or scale-based architectures prevalent in vision models [a,b]. It adopts a *channel-wise hierarchical organization*, where each view represents a distinct feature space. This design integrates DDCPs for dynamic reasoning adjustments based on context–target contrasts and MRM for progressive rule integration. Together, they form a *hierarchical reasoning framework* that explicitly mirrors compositional structure of latent rules. Our DSRF systematically enumerates rule combinations in a *principled and scalable* manner, enabling comprehensive coverage of complex attribute interactions while maintaining generalizability.
>
> - **DDCP enables dynamic reasoning.**
>
> DDCP enables dynamic reasoning behaviors by modeling context-target differences via gated attention guided by Gram matrices. This design allows the model to adapt its reasoning pathway based on specific context–target configuration of each sample, rather than relying on static mappings. DDCP learns dynamic mappings that continuously adapt to evolving context-target relationships, enabling robust generalization under distribution shift and novel rule configurations.
>
> - **DSRF: Novel** **dynamic and scalable reasoning framework**
>
> The integration of MVRP, DDCP and MRM forms a novel dynamic and scalable framework to reason for complex abstract rules. Moreover, it is adaptive and generalizable, as evidenced in OOD experiments on RAVENs [7, 8, 18], Unicode [9] and PGM [6] in Table 2-4, setting our approach apart from existing static or handcrafted reasoning strategies in abstract visual reasoning.
>
>
>
> We believe DSRF provides a principled and extensible framework for scaling up reasoning capacity without overfitting. We will emphasize these distinctions more clearly in the final version.
>
>
>
> [a] Lin, Tsung-Yi, et al. "Feature pyramid networks for object detection." CVPR, 2017.
>
> [b] Yang, Guoyu, et al. "Asymptotic feature pyramid network for labeling pixels and regions." TCSVT, 34(9), 2024: 7820-7829.
>
>
>
> **W2/Q2: Scalability analysis**
>
> Thank you for the insightful question.
>
> - Performance saturation in scaling DSRF
>
> The relatively small gain in Table 2, from 97.3% to 97.7%, is primarily due to the ceiling effect, where top-performing models all achieve high accuracy on RAVENs. Notably, DSRF-S significantly boosts the previously best-performing model, PredRNet [13], from 94.3% to 96.7%, while DSRF-M and DSRF-L further improve to 97.3% and 97.7%, all three significantly outperforming baselines despite the ceiling effect and demonstrating monotonically increasing gains by scaling.
>
> - Consistent performance gains by scaling up DSRFs
>
> Table A compares DSRF-S, DSRF-M and DSRF-L on all 6 RPM datasets. We added the held-out settings of PGM, *i.e.*, PGM-H.O.P (Held-out Attribute Pairs), PGM-H.O.TP (Held-out Triple Pairs), PGM-H.O.T (Held-out Triples), PGM-H.O.LT (Held-out Line-Type) and PGM-H.O.SC (Held-out Shape-Color) settings, each testing the model’s ability to generalize to novel combinations of attributes, unseen pairs or individual triples, and new attribute-object pairings.
>
> Scaling up DSRF yields consistent and substantial gains across all scenarios. Notably, under more challenging OOD settings on Unicode [9] and PGM [6], we observe more pronounced improvements, demonstrating that scalability becomes increasingly valuable when the reasoning task is more complex and the domain shift is substantial.
>
> |   |  | DSRF-S | DSRF-M | DSRF-L |
> | - | -| -| -| - |
> | RAVENs [7, 8, 18]  | O-RVN | 98.6   | 99.1   | 99.1   |
> || I-RVN   | 98.8 | 99.4   | 99.4   |
> || RVN-F   | 98.8  | 99.0   | 99.1   |
> | RAVENs OOD [7, 8, 18] | O-RVN -> I-RVN | 95.1   | 95.6   | 95.9   |
> || O-RVN -> RVN-F  | 98.5   | 98.7   | 98.9   |
> || I-RVN -> O-RVN  | 95.9  | 97.4   | 97.9   |
> || I-RVN -> RVN-F | 98.2   | 98.7   | 98.9   |
> || RVN-F -> O-RVN | 97.3 | 97.8   | 98.1   |
> || RVN-F -> I-RVN  | 95.4  | 95.8   | 96.5   |
> | Unicode [9]   |  | 72.1  | 73.2   | 73.8   |
> | Unicode OOD [9]  | | 34.5   | 35.9   | 37.3   |
> | PGM [6]    | PGM-N | 99.3   | 99.5   | 99.8   |
> || PGM-I    | 82.4  | 84.5   | 87.3   |
> || PGM-E    | 25.2 | 27.4   | 31.5   |
> || PGM-H.O.P    | 63.1 | 65.7   | 67.7   |
> || PGM-H.O.TP    | 75.7 | 78.8   | 82.3   |
> || PGM-H.O.T  | 29.3  | 31.3   | 34.2   |
> || PGM-H.O.LT    | 16.2  | 17.4   | 18.9   |
> || PGM-H.O.SC    | 16.1  | 16.7   | 17.8   |
> | RVP [37]     |    | 85.4   | 87.7   | 88.8   |
>
> Table A. Comparisons between DSRF-S, DSRF-M, DSRF-L on 6 RPM datasets.
>
>
>
> - Performance-complexity tradeoff analysis.
>
> While model scaling yields diminishing returns where near-optimal performance has already been achieved, it remains crucial for tackling harder OOD challenges. As evidenced above, scaled models exhibit enhanced capability in handling more complex cases, advancing toward developing more generalizable AI systems.
>
> - Practical meaning of scalability.
>
> Previous models typically scale reasoning capacity by stacking reasoning blocks, but generalize poorly [13, 17]. Our goal is not merely to increase model size, but more importantly to **expand reasoning capacity in a structured way while avoiding overfitting**. Specifically, the MVRP increases coverage of compositional rules by distributing reasoning across multiple attribute-based views. The DDCP adapts to varying rule distributions and focuses on relevant context-target differences. These mechanisms systematically scale reasoning capacity and effectively capture the latent rules present in training data while maintaining strong generalization on OOD samples, enabling the model to handle increasingly diverse and difficult reasoning scenarios.
>
>
>
> We will add more scalability analysis in the final manuscript.
>
>
>
> **W3/Q3: Unsubstantiated overfitting claim:**
>
> As noted in PredRNet [13] and HPAI [17], stacking reasoning modules often induces overfitting.
>
> | PredRNet [13]  | O-RVN | I-RVN | RVN-F |
> | - | - | - | - |
> |K=1  | 94.6 |  95.0 | 95.8  |
> | K=2  | 95.5 | 96.4  | 96.5  |
> | K=3  | 95.8 | 97.1  | 96.5  |
> | K=4  | 96.0 | 96.8  | 94.8  |
> |**HPAI [17]**  | **O-RVN** | **I-RVN** | **RVN-F**|
> | K=1 | 98.1 | 98.6  | 98.3  |
> | K=2 | 98.4 | 99.1  | 98.3  |
> | K=3 | 98.8 | 99.4  | 98.6  |
> | K=4 | 98.0 | 99.0  | 98.2  |
>
> Table B. Ablation of the number of reasoning blocks K in PredRNet [13] and HPAI [17].
>
> Both PredRNet [13] and HPAI [17] exhibit performance saturation or marginal degradation beyond three blocks, revealing inherent scalability constraints and depth-induced overfitting tendencies. These observations align with our motivation to pursue horizontal scalability through multi-view decomposition rather than relying on deep stacking.
>
> We will further clarify this in the final manuscript.
>
>
>
> **W4/Q4: Lack of comparison with LLMs:**
>
> In literature, researchers have applied LLMs for AVR tasks [c]. Table C compares DSRF-M with recent LLMs on RAVEN [18], reported by Apple [c]. LLMs exhibit near-chance performance on RAVEN, while dedicated AVR models like DSRF achieve almost perfect performance.
>
>
>
> |   | GPT-4V | Gemini Pro | QWen-VL-Max | LLaVA-1.5-13B | LLaVA-1.6-34B | DSRF-M |
> | - | -| - | - | - | - | -|
> | RAVEN [18] | 0.12  | 0.11| 0.10| 0.10  | 0.10 | 0.99   |
>
> Table C. Comparison to LLMs on RAVEN [18].
>
> [c] Zhang, Yizhe, et al. "How Far Are We from Intelligent Visual Deductive Reasoning?." ICLR 2024 Workshop: How Far Are We From AGI.
>
>
>
> **W5/Q5: Inconsistent structural presentation****.**
>
> We will revise the manuscript following your comments.
>
>
>
> **Q1: Each view in MVRP** **captures distinct attribute combinations****?**
>
> Thank you for the insightful question.
>
> - **Channel-wise decomposition facilitates feature disentanglement [b]**
>
> Although we do not impose explicit constraints such as orthogonality loss or entropy regularization, our model encourages view specialization implicitly through the multi-head view mapping in MVRP. Each view divides the global feature representation into attribute-relevant subspaces. Prior studies have shown that channel-wise decomposition can facilitate feature disentanglement and attribute grouping [d]. We allocate dedicated reasoning branch to each view, allowing different views to attend to different relational patterns. In addition, the subsequent DDCP modules adaptively extract localized reasoning patterns within each view, further reinforcing specialization across views during training.
>
> Following your comments, we ablate the case that we add an orthogonality loss as shown in Table D. This results in no performance improvement and may even lead to a slight accuracy drop due to minor conflicts between multiple loss terms.
>
> |  | RAVEN | I-RAVEN | RAVEN-FAIR |
> | - | - | - | - |
> | DSRF-M  | 99.1  | 99.4 | 99.0  |
> | +Orthogonality Loss | 98.9  | 99.2 | 98.9 |
>
> Table D. Comparison to adding orthogonality loss.
>
>
>
> - **Clustering analysis**
>
> Since figures can’t be reported during rebuttal, we compute cosine similarity between the four views' feature maps in DSRF-M's first layer, following [e]. The results in Table E indicate low interview similarity, supporting the emergence of view-wise specialization.
>
> | Sim(v1, v2) | Sim(v1, v3) | Sim(v1, v4) | Sim(v2, v3) | Sim(v2, v4) | Sim(v3, v4) |
> | - | - | - | - | -| - |
> | -0.06 | 0.06  | -0.08  | 0.03 | -0.03 | 0.01 |
>
> Table E. Feature similarity across views.
>
> We will include t-SNE clustering analysis in the final manuscript to show that each view captures different aspects of the input.
>
> [d] Hu, J., Shen, L., & Sun, G. Squeeze-and-excitation networks. CVPR, 2018, pp. 7132–7141.
>
> [e] Zhao, Wenliang, et al. "Towards interpretable deep metric learning with structural matching." ICCV, 2021.

---

> ### Comment · Reviewer_Yeks · 2025-08-06
>
> The author's response has resolved most of my concerns, and I recommend increasing their score.

---

> > ### Author Response · Authors · 2025-08-07
> >
> > We sincerely appreciate your positive feedback and the time you have taken to review our rebuttal. We are grateful for your thoughtful comments and would be pleased to address any remaining concerns you may have.

---

### Official Review · Reviewer_UaAG · 2025-07-01

**Clarity:** 3
**Significance:** 2
**Originality:** 3
**Rating:** 4
**Confidence:** 3

**Summary:**

The core breakthrough of DSRF is to solve the static architecture defects of HP²AI and other models through width expansion (MVRP) and dynamic domain adaptation (DDCP), and significantly improve the performance in OOD scenarios and complex rule tasks, but at a higher computational cost.

**Questions:**

1. In Table 4 and Table 5, how does the DSRF-S model perform?
2. In ablation experiments, how much accuracy loss will be caused by the alternative scheme of DDCP Gram matrix (such as covariance matrix) ?

**Ethical Concerns:**

["NO or VERY MINOR ethics concerns only"]

**Limitations:**

Yes.

**Paper Formatting Concerns:**

None.

**Quality:**

3

**Strengths And Weaknesses:**

Strength

1. The experiments contains many baselines and datasets.

2. The author analyzes the failed cases in the appendix.

Weakness

1. Although many models have been compared in the experiment, there are few models (10M~25M) with the same size as DSRF-S or DSRF-M as the baseline. Therefore, it is difficult to conclude that the improvement of model capability is due to the author's special model design method or the growth of model parameters.

2. In terms of computing efficiency, it is a pity that the lightweight scheme was not discussed.

---

> ### Author Rebuttal · Authors · 2025-07-31
>
> **W1: Performance gain from model design or model growth****?**
>
> Thank you for your insightful comments. The two principal design objectives of this work, as emphasized in the title, are dynamic adaptability and scalability. Specifically, we aim to develop a scalable abstract visual reasoning framework while effectively tackling the challenge of overfitting.
>
> - **Simply stacking reasoning modules would lead to overfitting**
>
> As noted in the original PredRNet [13] and HPAI [17] studies, stacking deep reasoning modules frequently induces overfitting, despite aiming to enhance reasoning through additional blocks.
>
> | PredRNet [13]     | O-RVN     | I-RVN     | RVN-F     |
> | ----------------- | --------- | --------- | --------- |
> | K=1               | 94.6      | 95.0      | 95.8      |
> | K=2               | 95.5      | 96.4      | 96.5      |
> | K=3               | 95.8      | 97.1      | 96.5      |
> | K=4               | 96.0      | 96.8      | 94.8      |
> | **HPAI** **[17]** | **O-RVN** | **I-RVN** | **RVN-F** |
> | K=1               | 98.1      | 98.6      | 98.3      |
> | K=2               | 98.4      | 99.1      | 98.3      |
> | K=3               | 98.8      | 99.4      | 98.6      |
> | K=4               | 98.0      | 99.0      | 98.2      |
>
> Table A. Experimental results of stacking K reasoning blocks in PredRNet [13] and HPAI [17].
>
> As shown above, both PredRNet [13] and HPAI [17] exhibit performance saturation or marginal degradation beyond three blocks, revealing inherent scalability constraints and depth-induced overfitting tendencies. These observations align with our motivation to pursue horizontal scalability through multi-view decomposition rather than relying on deeper stacking.
>
> - **Our scaled-up models maintain generalization**
>
> As shown in Tables 1–3, scaled-up models (DSRF-M/L) consistently outperform smaller variants (DSRF-S/M) on both RAVEN [7,8,18] and Unicode [9] benchmarks. We acknowledge that DSRF falls within a relatively large parameter range (10M–40M), but our scaled-up architectures successfully learn latent rule representations while preserving robust generalization capabilities across diverse task distributions.
>
> This robustness is attributed to the novel architecture of MVRP, DDCP and MRM. Each view is processed independently through modular DDCP–MRM branches, which mitigates overfitting by encouraging localized, attribute-sensitive reasoning. The DDCP block promotes relational comparisons instead of direct memorization, while the MVRP introduces compositional constraints that prevent the model from relying on shortcut patterns. Together, these mechanisms guide the model to learn transferable rule structures, allowing larger variants to benefit from increased capacity without overfitting.
>
> - **Ablation shows the effectiveness of designed modules.**
>
> To assess the individual contribution of our core modules, we provide detailed ablation studies in Table 6. The ablations show that removing designed components leads to substantial performance degradation, even under the same model scale. In addition, in Table 7, we show that replacing the reasoning module of state-of-the-art models by DSRF-S and DSRF-M significantly improves the reasoning accuracy. All these ablations support our claim that the improvement of performance comes from the proposed design rather than simply growing models.
>
>
>
> We will further clarify this point in the final manuscript and provide more justifications.
>
>
>
> **W2: Lightweight scheme of DSRF**
>
> Thank you for your insightful comments. Although the original manuscript did not specifically investigate lightweight implementations, our framework inherently supports efficient deployment through its modular architecture, enabling straightforward adaptation to resource-constrained scenarios.
>
> - **Lightweight scheme by adjusting MVRP hyperparameters.**
>
> In particular, the MVRP allows for controlling the channel dimensions of feature mappings within each view. By decreasing the dimensionality of each mapped view, the model can be scaled down to more lightweight variants with lower memory and FLOPs, while still maintaining core reasoning capabilities.
>
> - **Results of lightweight DSRFs**
>
> By reducing the channel dimensions of each view by half, we can construct a lightweight variant of DSRF-S and DSRF-M, denoted as DSRF-S-Light and DSRF-M-Light. We conduct comparison experiments on three RAVEN datasets and three PGM settings (PGM-N, PGM-I, PGM-E). The lightweight variants achieve substantial reductions in model size and computational cost (GFLOPs) while maintaining reasonable accuracy.
>
>
>
> |              | O-RVN | I-RVN | RVN-F | PGM-N | PGM-I | PGM-E | GFLOPs | Model Size (M) |
> | ------------ | ----- | ----- | ----- | ----- | ----- | ----- | ------ | -------------- |
> | DSRF-S-Light | 98.5  | 98.5  | 98.6  | 99.1  | 81.7  | 24.8  | 4.06   | 4.96           |
> | DSRF-S       | 98.7  | 98.6  | 98.8  | 99.3  | 82.4  | 25.2  | 10.10  | 11.64          |
> | DSRF-M-Light | 98.8  | 98.9  | 98.8  | 99.3  | 83.1  | 26.1  | 5.11   | 8.72           |
> | DSRF-M       | 99.1  | 99.4  | 99.0  | 99.5  | 84.5  | 27.4  | 13.92  | 23.86          |
>
> Table B. Comparisons with lightweight DSRFs.
>
>
>
> We will supplement the lightweight schemes and related experimental results in the final manuscript and the supplementary materials accompanying the publication.
>
>
>
> **Q1**: **In Table 4 and Table 5, how does the DSRF-S model perform?**
>
> We supplement the results of DSRF-S as follows.
>
> |        | PGM-N | PGM-I | PGM-E | RVP  |
> | ------ | ----- | ----- | ----- | ---- |
> | DSRF-S | 99.3  | 82.4  | 25.2  | 85.4 |
> | DSRF-M | 99.5  | 84.5  | 27.4  | 87.7 |
>
> Table C. Evaluation of DSRF-S and DSRF-M on PGM [6] and RVP [37] datasets.
>
>
>
> While DSRF-S demonstrates inferior performance relative to DSRF-M, it maintains competitive advantages over alternative baseline models, consistent with our expectations.
>
>
>
> **Q2: Performance loss by replacing DDCP Gram matrix** **with an alternative scheme**
>
> Thank you for the question. We conduct additional ablation experiments where the Gram matrix in DDCP is replaced with a covariance matrix. This modification results in a performance drop of approximately 0.4% on three RAVEN datasets and 0.6% on three PGM settings (PGM-N, PGM-I, PGM-E), indicating that the Gram matrix is more effective than the covariance matrix.
>
>
>
> |                 | O-RVN | I-RVN | RVN-F | PGM-N | PGM-I | PGM-E |
> | --------------- | ----- | ----- | ----- | ----- | ----- | ----- |
> | DDCP-Gram       | 99.1  | 99.4  | 99.0  | 99.5  | 84.5  | 27.4  |
> | DDCP-Covariance | 98.8  | 99.0  | 98.6  | 99.4  | 83.6  | 26.8  |
>
> Table D Comparisons between Gram matrix and covariance matrix in DDCP.
>
>
>
> We will supplement this ablation in the supplementary materials accompanying the publication.

---

### Official Review · Reviewer_UTPG · 2025-07-01

**Clarity:** 2
**Significance:** 3
**Originality:** 2
**Rating:** 4
**Confidence:** 4

**Summary:**

This paper proposes DSRF, a new abstract visual reasoning framework to address the limitations of scalability and adaptability in existing models. The proposed MVRP module maps visual features into multiple views to capture complex reasoning rules. The DDCP module introduces a dynamic attention mechanism, enabling the reasoning network to adaptively handle abstract rules in various tasks. Experiments conducted on six AVR datasets demonstrate that DSRF achieves SOTA accuracy on these datasets, exhibiting outperforming abstract visual reasoning ability.

**Questions:**

1. Has DSRF been evaluated under held-out settings of PGM or RAVEN?

2. Can we apply DSRF to non-matrix abstract visual reasoning tasks, such as the Bongard Problems?

**Ethical Concerns:**

["NO or VERY MINOR ethics concerns only"]

**Final Justification:**

The authors' responses have addressed my concerns about computational efficiency and OOD evaluation. They have reported additional results to show the model's performance on OOD configurations. So I maintained the original score.

**Limitations:**

yes

**Paper Formatting Concerns:**

I think there is no formatting issue that needs to be discussed here.

**Quality:**

3

**Strengths And Weaknesses:**

Strengths

The paper proposes novel modules with both scalability and adaptability. MVRP expands the space of representable abstract rules by constructing multiple views from representations, allowing it to model a wide range of compositional rules. DDCP can dynamically attend to domain differences in flexibly adapt to task-specific abstract rules. The experimental section is thorough, and the results show that DSRF achieves SOTA performance across the six abstract visual reasoning datasets.

Weaknesses

1. While the proposed modules enhance model capacity, they may introduce additional computational overhead. The paper does not provide a quantitative analysis of model efficiency, such as inference speed or memory consumption, nor does it compare the number of model parameters across the compared models.

2. The data efficiency and training data requirements of DSRF are not discussed. A discussion on how the model performs under limited training data would be valuable.

3. There is ambiguity in the paper’s definition of OOD evaluation. If the differences among I-RVN, O-RVN, and RVN-F lie solely in the provided options, while the attributes and rules remain unchanged, then these settings may not constitute true OOD tests. Since existing methods have achieved strong performance on datasets like RAVEN, the performance of DSRF on unseen rule-attribute combinations is critical.

4. The definitions of Θp and Θr are missing in Line 107.

---

> ### Author Rebuttal · Authors · 2025-07-31
>
> **W1: Computational complexity Analysis**
>
> The reviewer may overlook the model complexity analysis in the last two columns of Table 2, and the associated discussions at Line 233-236 in Sec. 4.2, where we reported GFLOPs and model size (M) of compared models. One of our key contributions is to introduce a scalable design to boost the reasoning capability of our models. As shown in Table 2, our DSRF-S, DSRF-M and DSRF-L outperform all the compared models. Through systematic model scaling, we demonstrate progressive improvements in reasoning capability across the DSRF architecture variants (S→M→L), validating the effectiveness of our scalable design paradigm.
>
>
>
> **W2: Data efficiency**
>
> Thank you for your comments. Following your comments, we conducted experiments on the SVRT dataset using 500, 1k, 5k, and 10k training samples respectively, following GAMR [b]. This evaluation provides a controlled setting to assess generalization under limited data. The results are summarized below (left: same-different tasks/right: spatial related tasks). DSRF achieves strong performance across various SVRT tasks, even when trained with limited samples such as 500 samples per task. The results demonstrate its ability to learn abstract visual rules efficiently.
>
>
>
> |          | ResNet-50 [c] | RN [6] | ESBN [a] | Atten-ResNet [d] | GAMR [b] | DSRF-M |
> | -------- | ------------- | ------ | -------- | ---------------- | -------- | ------ |
> | SVRT-500 | 54/85         | 53/85  | 58/64    | 63/95            | 78/98    | 86/100 |
> | SVRT-1k  | 57/96         | 58/90  | 58/64    | 69/98            | 83/99    | 92/100 |
> | SVRT-5k  | 78/100        | 68/97  | 63/68    | 88/100           | 94/100   | 98/100 |
> | SVRT-10k | 84/100        | 75/97  | 67/67    | 92/100           | 95/100   | 99/100 |
>
> Table A. Sample efficiency analysis on the SVRT dataset.
>
>
>
> We will add this data efficiency analysis into the in the final manuscript and the supplementary materials accompanying the publication.
>
>
>
> [a] Webb, T. W., Sinha, I., and Cohen, J. D. Emergent symbols through binding in external memory. In ICLR, 2021.
>
> [b] Vaishnav, M. and Serre, T. Gamr: A guided attention model for (visual) reasoning. In ICLR, 2023.
>
> [c] He, Kaiming, et al. "Deep residual learning for image recognition." *CVPR,* 2016.
>
> [d] Vaishnav, Mohit, et al. "Understanding the computational demands underlying visual reasoning." *Neural Computation* 34.5 (2022): 1075-1099.
>
>
>
> **W3: Ambiguity in OOD evaluation**
>
> Thank you for your comments. We would like to clarify that OOD (Out-of-Distribution) may refer to different levels of distribution shift.
>
> - **Distribution shifts due to novel rule-attribute combinations on PGM and Unicode datasets****.**
>
> - We have conducted experiments on OOD settings on Unicode Analogy [9], where all the symbols in the training set do not appear in the test set. The results are reported in Table 3 and the related discussions can be found at Line 237-250.
>
> - In addition, we have conducted experiments on OOD settings on PGM [6], Interpolation that evaluates generalization to intermediate attribute values, and Extrapolation that tests the model’s ability to reason beyond the observed attribute range. The results are reported in Table 4 and the related discussions can be found at Line 251-260.
>
> - Furthermore, we provide the experimental results on held-out settings of PGM in the table below, *i.e.*, PGM-H.O.P (Held-out Attribute Pairs), PGM-H.O.TP (Held-out Triple Pairs), PGM-H.O.T (Held-out Triples), PGM-H.O.LT (Held-out Line-Type) and PGM-H.O.SC (Held-out Shape-Color) settings, each testing the model’s ability to generalize to novel combinations of attributes, unseen pairs or individual triples, and new attribute-object pairings absent during training. Our models perform best across almost all settings.
>
> |                     | PGM-N | PGM-I | PGM-E | PGM-H.O.P | PGM-H.O.TP | PGM-H.O.T | PGM-H.O.LT | PGM-H.O.SC |
> | ------------------- | ----- | ----- | ----- | --------- | ---------- | --------- | ---------- | ---------- |
> | ARII [a]            | 88.0  | 72.0  | 29.0  | 50.0      | 64.1       | 32.1      | 16.0       | 12.7       |
> | Slot-Abstractor [b] | 91.5  | 91.6  | 39.3  | 63.3      | 78.3       | 20.4      | 16.7       | 14.3       |
> | DSRF-M              | 99.5  | 84.5  | 27.4  | 65.7      | 78.8       | 31.3      | 17.4       | 16.7       |
> | DSRF-L              | 99.8  | 87.3  | 31.5  | 67.7      | 82.3       | 34.2      | 18.9       | 17.8       |
>
> Table B. Comparisons on PGM OOD regimes.
>
>
>
> - Distribution shifts due to novel samples on RAVENs
>
> The OOD settings across I-RVN, O-RVN, and RVN-FAIR involve shifts in image distributions and answer choice generation strategies, testing novel samples while the underlying rule-attribute combinations remain largely consistent. Under these OOD settings, our DSRFs significantly outperforms compared methods, as shown in Table 2.
>
>
>
> We acknowledge that the OOD settings in PGM [6] and Unicode [9] are more challenging than OOD settings on RAVENs. Nevertheless, our model demonstrates strong performance under these OOD regimes, as reported in Tables 3 and 4, showing that DSRF is capable of generalizing beyond memorized rule structures and adapting to new reasoning patterns.
>
>
>
> We acknowledge the current ambiguity in defining out-of-distribution (OOD) evaluation settings. For the final version, we will rigorously specify the OOD criteria for each benchmark (RAVENs [7,8,18], PGM [6], and Unicode [9]), explicitly delineating their distinct characteristics.
>
>
>
> **Q1: Has DSRF been evaluated under held-out settings of PGM or RAVEN?**
>
> In the original manuscript, we evaluate DSRF under two OOD settings of the PGM dataset: Interpolation and Extrapolation. Following your suggestion, we have additionally provided results on **held-out settings of PGM** [6] as shown in the reply to W3 above. To our best knowledge, no held-out settings were defined for RAVENs [7, 8, 18].
>
>
>
> **Q2: Non-matrix abstract visual reasoning such as Bongard problem**
>
> Yes, the core components of DSRF are general and can be adapted to other forms of abstract visual reasoning, including non-matrix tasks like the Bongard problems.
>
> - **Adaptation to Bongard Problems**
>
> Specifically, the DDCP block only requires a set of context-target pairs and does not rely on any fixed spatial layout. Similarly, the MVRP can be applied to grouped features based on visual attributes rather than grid positions. To adapt DSRF to Bongard problems, one may treat the positive or negative example sets as a separate context-to-target reasoning branch and apply DDCP to extract the reasoning patterns.
>
> - **Evaluation of** **Bongard problems**
>
> Following your comments, we add the experimental results of DSRF on Bongard-OpenWorld [c].
>
>
>
> |      | GPT-4 [d] | SNAIL [e] | DSRF-S | DSRF-M | DSRF-L |
> | ---- | --------- | --------- | ------ | ------ | ------ |
> | Acc. | 63.3      | 64.0      | 64.3   | 65.2   | 66.1   |
>
> **Table C.** Comparisons on **Bongard problems****.**
>
>
>
> The results of GPT-4 [d] and SNAIL [e] are obtained from [c]. We evaluate DSRFs following the same evaluation protocol [c]. DSRFs outperform the compared methods on Bongard problems, even the large language model like GPT-4 [d].
>
>
>
> We will update the results and the related discussions in the final manuscript.
>
>
>
> [c] Wu, Rujie, et al. "Bongard-openworld: Few-shot reasoning for free-form visual concepts in the real world." arXiv preprint arXiv:2310.10207 (2023).
>
> [d] Bubeck, Sébastien, et al. "Sparks of artificial general intelligence: Early experiments with gpt-4." arXiv preprint arXiv:2303.12712 (2023).
>
> [e] Mishra, Nikhil, et al. "A Simple Neural Attentive Meta-Learner." ICLR 2018.

---

> > ### Comment · Reviewer_UTPG · 2025-08-08
> >
> > Thank you for the detailed response. These results have addressed most of my earlier concerns. But I fint that DSRF’s performance advantage appears to be less pronounced in the OOD setting. Could the authors explain the reason for this?

---

> > > ### Author Response · Authors · 2025-08-09
> > >
> > > Thank you for your positive feedback. We would like to respond to your further question on the less pronounced gains on OOD settings as follow.
> > >
> > > **OOD settings challenge models’ generalization capabilities.**
> > >
> > > The intrinsic generalization difficulties inherent in OOD scenarios often exacerbate the overfitting propensity of large-scale models, as their high-dimensional parameter spaces predispose them to over-optimize toward domain-specific idiosyncrasies while compromising the learning of transferable feature representations. Although the proposed DSRF framework successfully mitigates overfitting through its DDCP module, OOD conditions continue to present substantial challenges, markedly degrading model performance. This effect is particularly pronounced in larger model architectures (DSRF-M/L), which face heightened demands on their generalization capabilities. However, the DDCP module's dynamic reasoning mechanisms enable these scaled-up DSRF variants to achieve consistent performance gains despite these adversarial conditions.
> > >
> > > **Consistent performance gains by scaling up DSRFs**
> > >
> > > Tab. A summarizes comparisons with previous best models and our DSRFs. Our DSRFs perform best in most OOD regimes and scaled-up models like DSRF-M and DSRF-L achieve consistent performance gains. Note that previous best models are different in different settings. We will include these results in the final version.
> > >
> > > | | | Previous  Best | DSRF-S | DSRF-M | DSRF-L |
> > > |-|-|-|-|-|-|
> > > | RAVENs [7, 8, 18]| O-RVN| 98.8| 98.6| 99.1| 99.1 |
> > > | | I-RVN | 99.4 | 98.8 | 99.4 | 99.4 |
> > > | | RVN-F | 98.6 | 98.8   | 99.0 | 99.1 |
> > > | RAVENs  OOD [7, 8, 18] | O-RVN->I-RVN | 89.2 | 95.1 | 95.6   | 95.9   |
> > > | | O-RVN->RVN-F | 97.9 | 98.5 | 98.7 | 98.9 |
> > > | | I-RVN->O-RVN | 93.9 | 95.9 | 97.4 | 97.9 |
> > > | | I-RVN->RVN-F | 97.3  | 98.2 | 98.7 | 98.9|
> > > | | RVN-F->O-RVN | 96.2 | 97.3| 97.8 | 98.1|
> > > | | RVN-F->I-RVN | 91.4 | 95.4| 95.8  | 96.5|
> > > | Unicode  [9] | | 57.7 | 72.1| 73.2 | 73.8|
> > > | Unicode  OOD [9] | | 30.2 | 34.5 | 35.9 | 37.3|
> > > | PGM  [6] | PGM-N  | 99.3  | 99.3 | 99.5 | 99.8|
> > > | | PGM-I  | 91.6 | 82.4| 84.5 | 87.3|
> > > | | PGM-E  | 39.3 | 25.2| 27.4 | 31.5|
> > > | | PGM-H.O.P | 63.3 | 63.1 | 65.7| 67.7   |
> > > | | PGM-H.O.TP | 78.3| 75.7 | 78.8| 82.3   |
> > > | | PGM-H.O.T  | 32.1| 29.3 | 31.3| 34.2   |
> > > | | PGM-H.O.LT   | 16.7| 16.2 | 17.4| 18.9   |
> > > | | PGM-H.O.SC   | 14.3 | 16.1 | 16.7   | 17.8   |
> > > | RVP [37]|  | 71.6 | 85.4   | 87.7   | 88.8   |
> > >
> > > Table A. Comparisons between previously best performing models, DSRF-S/M/L on 6 RPM datasets.
> > >
> > > **DDCP enables robust generalization through dynamic reasoning.**
> > >
> > > While scaled-up models tend to overfit, DDCP enables dynamic reasoning behaviors by modeling context-target differences via gated attention guided by Gram matrices. This design allows the model to adapt its reasoning pathway based on specific context–target configuration of each sample, rather than relying on static mappings. DDCP learns dynamic mappings that continuously adapt to evolving context-target relationships, enabling robust generalization under distribution shift and novel rule configurations.
> > >
> > > **Row-wise reasoning focus and column-wise extension.**
> > >
> > > DSRF initially focuses on row-wise rule learning, as most RPM datasets [7, 8, 18] primarily employ row-wise patterns. This design prioritizes efficiency while maintaining effectiveness for standard RPM tasks. However, this approach shows limitations on PGM [6], which specifically emphasizes column-wise relational reasoning.
> > >
> > > While DSRF doesn't explicitly model column-wise rules, the shared-weight architecture across rows enables implicit column-wise pattern recognition. The uniform application of DDCP modules across rows encourages learning generalizable features that may naturally extend to column-wise regularities.
> > >
> > > To explicitly enhance this capability, we introduce a column-wise DDCP branch, fusing its outputs with row-wise features. The modular DDCP design facilitates this seamless extension. Our experiments demonstrate substantial performance improvements (see DSRF-M/L-RC results), with greater gains over baselines than row-wise-only variants. Complete implementation details and these results will be provided in the final version.
> > >
> > > | | PGM-N | PGM-I  | PGM-E | PGM-H.O.P | PGM-H.O.TP | PGM-H.O.T | PGM-H.O.LT | PGM-H.O.SC |
> > > | -| -| -| - | -| - | - | - | - |
> > > | ARII [a] | 88.0 | 72.0  | 29.0 | 50.0 | 64.1 | 32.1 | 16.0 | 12.7 |
> > > | Slot-Abstractor  [b] | 91.5 | 91.6| 39.3 | 63.3 | 78.3 | 20.4 |16.7 |14.3|
> > > | DSRF-M | 99.5 | 84.5 | 27.4| 65.7| 78.8| 31.3 | 17.4| 16.7|
> > > | DSRF-M-RC | 99.7 | 88.9| 35.1| 68.9 | 82.1 | 34.1| 19.1| 18.1|
> > > | DSRF-L  | 99.8 | 87.3 | 31.5 | 67.7| 82.3 | 34.2| 18.9| 17.8|
> > > | DSRF-L-RC | **99.9** | **92.1** | **40.2** | **71.3**  | **86.7**   | **37.3**  | **19.9**   | **19.4**   |
> > >
> > > Table B: Comparisons on PGM.
> > >
> > > [a] Learning robust rule representations for abstract reasoning via internal inferences. NeurIPS 2022.
> > >
> > > [b] Slot Abstractors: Toward Scalable Abstract Visual Reasoning. ICML 2024.

---

### Official Review · Reviewer_Bup5 · 2025-07-03

**Clarity:** 3
**Significance:** 2
**Originality:** 3
**Rating:** 4
**Confidence:** 3

**Summary:**

The paper proposes a new reasoning framework, called dynamic and scalable reasoning framework (DSRF) for solving abstract visual reasoning problems, specially Raven’s Progressive matrices (RPMs). DSRF consists of blocks like multi-view reasoning pyramid to widen the network to capture more attribute combinations, dynamic domain contrast prediction block to dynamically capture rules with the help of a gated attention reasoning block. Finally a multi-granularity rule mixing block is proposed to capture different combinations of rules from different perspectives. Through extensive results on six RPM benchmarks, including one involving real world traffic prediction, the authors demonstrate the effectiveness of the framework. The authors also demonstrate that DSRF outperforms other methods under different OOD settings and can be used to replace the reasoning module of other models.

**Questions:**

1. How sample-efficient is DSRF? Can the authors evaluate on ART [1] /SVRT [2] tasks, where sample efficiency can be systematically evaluated?

2. Is the model trained using the standard scoring mechanism used in RPMs, where for each answer choice a score is obtained which is used to compute cross entropy loss?

3. How important is the multi-view reasoning pyramid to the overall performance?

4. Given that the MRM module seems the least significant component, its not clear to me how the different mixings are important. Perhaps some ablations targeting those could be helpful and also some additional information about how these mixings are done.

[1] - Webb, T. W., Sinha, I., and Cohen, J. D. Emergent symbols through binding in external memory. In 9th International Conference on Learning Representations, ICLR, 2021.

[2] - Vaishnav, M. and Serre, T. Gamr: A guided attention model for (visual) reasoning. In 11th International Conference on Learning Representations, ICLR, 2023

**Ethical Concerns:**

["NO or VERY MINOR ethics concerns only"]

**Final Justification:**

The authors' rebuttal has addressed my concerns and questions, so I raised my score.

**Limitations:**

Discussed in appendix, would be good to have a summarized version in the main paper.

**Quality:**

2

**Strengths And Weaknesses:**

Strengths:

1. The paper is well written.
2. Proposed a new reasoning framework DSRF for RPMs, consisting of blocks like multi-view reasoning pyramid, dynamic domain contrast prediction, multi-granularity rule mixer targeting different aspects relevant for abstract visual reasoning.
3. Extensive results on six benchmarks (including real-world images) demonstrate its effectiveness.
DSRF can also be used to replace the reasoning frameworks of other models, to further improve their performance on RPM benchmarks.


Weaknesses:

1. The authors don’t compare with ARII [1] and Slot-Abstractor [2] for the PGM which especially outperforms other methods on OOD generalization regimes, and it seems both of them achieve higher scores than DSRF on extrapolation. Also it would be valuable to evaluate DSRF on other OOD generalization regimes in the PGM dataset like Heldout shape color, Heldout line type, Heldout triples etc some of which are more difficult than the extrapolation regime.

2. Given that operations within DSRF extract rules by processing the RPMs rowwise, it seems this framework won’t be able to capture any rules columnwise.

[1] - Zhang, W., Liu, X. and Song, S., 2022. Learning robust rule representations for abstract reasoning via internal inferences. Advances in Neural Information Processing Systems, 35, pp.33550-33562

[2] - Mondal, S.S., Cohen, J.D. and Webb, T.W., 2024. Slot abstractors: Toward scalable abstract visual reasoning. arXiv preprint arXiv:2403.03458

---

> ### Author Rebuttal · Authors · 2025-07-31
>
> **W1. Comparison on PGM OOD regimes**
>
> Following your comments, we have evaluated DSRF-M and DSRF-L on all PGM tasks, including PGM-H.O.P (Held-out Attribute Pairs), PGM-H.O.TP (Held-out Triple Pairs), PGM-H.O.T (Held-out Triples), PGM-H.O.LT (Held-out Line-Type) and PGM-H.O.SC (Held-out Shape-Color) settings, each testing the model’s ability to generalize to novel combinations of attributes, unseen pairs or individual triples, and new attribute-object pairings.
>
> | | PGM-N | PGM-I | PGM-E | PGM-H.O.P | PGM-H.O.TP | PGM-H.O.T | PGM-H.O.LT | PGM-H.O.SC |
> | -| - | - | -| - | -| -| -| -|
> | ARII [a] | 88.0  | 72.0  | 29.0  | 50.0 | 64.1| 32.1| 16.0 | 12.7 |
> | Slot-Abstractor [b] | 91.5  | 91.6  | 39.3  | 63.3  | 78.3  | 20.4 | 16.7 | 14.3  |
> | DSRF-M | 99.5  | 84.5  | 27.4  | 65.7 | 78.8 | 31.3  | 17.4  | 16.7  |
> | DSRF-L | 99.8  | 87.3  | 31.5  | 67.7 | 82.3 | 34.2  | 18.9       | 17.8   |
>
> Table A. Comparisons on PGM OOD regimes.
>
>
> As noted by the reviewer, DSRF underperforms Slot-Abstractor [b] in Interpolation and Extrapolation tasks. This is mainly because DSRF adopts a more general reasoning paradigm without accessing attribute labels, while Slot-Abstractor [b] utilizes these attribute labels to model continuous attribute variations effectively.
>
> On other OOD settings, DSRF consistently outperforms ARII [a] and Slot-Abstractor [b]. This is mainly because these settings involve more complex rule-attribute compositions, where DSRF's dynamic and modular design enables stronger generalization beyond fixed attribute-slot mappings.
>
> DSRF-L surpasses ARII [a] in all PGM tasks and Slot-Abstractor [b] in 6/8 tasks, while DSRF-M outperforms ARII [a] in 7/8 tasks and Slot-Abstractor [b] in 6/8 tasks. These results demonstrate their superior performance in PGM OOD tasks.
>
> We will include these experimental results in the final manuscript.
>
>
>
> **W2: Handling column-wise rule**
>
> Thank you for your insightful comments. We respond to your comments as follows.
>
> - Intuitions for modelling row-wise rules only.
>
> DSRF focuses on row-wise rules only as most datasets such as RAVEN [18], I-RAVEN [7], RAVEN-FAIR [8] only implement row-wise rules. We prioritized efficiency by focusing on row-wise reasoning, which suffices for most RPM tasks but admittedly limits the model.
>
> - Not explicitly but implicitly model column-wise rules.
>
> - While DSRF does not explicitly model column-wise rules, column-wise reasoning emerges implicitly through the shared weight architecture across rows. As the same DDCP module is applied to all rows, it encourages the model to learn generalizable patterns that may also align along columns, allowing the model to partially capture column-wise regularities.
>
> - In all the datasets we evaluated, only PGM [6] implements column-wise rules. By utilizing row-wise DDCP only, DSRF performs excellently on PGM, as shown in Table 4.
>
> - Incorporating column-wise DDCP.
>
> Following your comments, we extend the framework by adding a column-wise DDCP reasoning branch, whose outputs are fused with the row-wise DDCP to enhance rule representation. The DDCP’s modular design enables easy adaptation for column-wise processing.
>
> We evaluate this extension on PGM as shown below, where DSRF-M-RC/DSRF-L-RC denote the DSRF-M/DSRF-L variants utilizing both row-wise and column-wise DDCPs. By adding column-wise DDCPs, the performance significantly improves.
>
>
>
> |           | PGM-N | PGM-I | PGM-E | PGM-H.O.P | PGM-H.O.TP | PGM-H.O.T | PGM-H.O.LT | PGM-H.O.SC |
> | -- | - | --| ----- | --------- | ---------- | --------- | ---------- | ---------- |
> | DSRF-M    | 99.5  | 84.5  | 27.4  | 65.7      | 78.8       | 31.3      | 17.4       | 16.7       |
> | DSRF-M-RC | 99.7  | 88.9  | 35.1  | 68.9      | 82.1       | 34.1      | 19.1       | 18.1       |
> | DSRF-L    | 99.8  | 87.3  | 31.5  | 67.7      | 82.3       | 34.2      | 18.9       | 17.8       |
> | DSRF-L-RC | 99.9  | 92.1  | 40.2  | 71.3      | 86.7       | 37.3      | 19.9       | 19.4       |
>
> Table B: Results for DSRF-M-RC and DSRF-L-RC on PGM.
>
>
>
> We will revise these results and related discussions in the final manuscript.
>
>
>
> [a] Webb, T. W., Sinha, I., and Cohen, J. D. Emergent symbols through binding in external memory. In ICLR, 2021.
>
> [b] Vaishnav, M. and Serre, T. GAMR: A guided attention model for (visual) reasoning. In ICLR, 2023.
>
>
>
> **Q1: Evaluating** **sample-efficient of DSRF** **on SVRT**
>
> Following your comments, we conducted experiments on the SVRT dataset using 500, 1k, 5k, and 10k training samples respectively, following GAMR [b]. This evaluation provides a controlled setting to assess generalization under limited data. The results are summarized below (left: same-different tasks/right: spatial related tasks). DSRF achieves strong performance across various SVRT tasks, even when trained with limited samples such as 500 samples per task. The results demonstrate its ability to learn abstract visual rules efficiently.
>
> |   | ResNet-50 [c] | RN [6] | ESBN [a] | Atten-ResNet [d] | GAMR [b] | DSRF-M |
> | - | - | - | - | - | - | - |
> | SVRT-500 | 54/85 | 53/85  | 58/64    | 63/95  | 78/98  | 86/100 |
> | SVRT-1k  | 57/96  | 58/90  | 58/64    | 69/98    | 83/99  | 92/100 |
> | SVRT-5k  | 78/100  | 68/97  | 63/68    | 88/100  | 94/100 | 98/100 |
> | SVRT-10k | 84/100  | 75/97  | 67/67    | 92/100  | 95/100 | 99/100 |
>
> Table C. Sample efficiency analysis on the SVRT dataset.
>
>
>
> We will add this sample efficiency analysis into the in the final manuscript and the supplementary materials accompanying the publication.
>
>
>
> [c] He, Kaiming, et al. "Deep residual learning for image recognition." *CVPR*, 2016.
>
> [d] Vaishnav, Mohit, et al. "Understanding the computational demands underlying visual reasoning." *Neural Computation* 34.5 (2022): 1075-1099.
>
>
>
> **Q2: Standard scoring mechanisms****?**
>
> Yes, our model uses the standard RPM scoring mechanism, consistent with PredRNet [13] and many other models [12, 17]. We apply binary cross-entropy loss over the scores of the eight answer choices, which is functionally equivalent to Softmax with cross-entropy loss under the condition of single-choice questions.
>
>
>
> **Q3: Importance of multi-view reasoning pyramid**
>
> Multi-view reasoning pyramid serves as the foundational architecture of our framework, enabling its scalable design and robust performance. By utilizing different number of views, we design three pyramids with small, medium and large configurations, with a pyramid structure of “2-1”, “4-2-1”, and “8-4-1”, where each number represents the number of views in each layer. Combined with the dynamic reasoning capability enabled by DDCP and its core module, GARB, our DSRF demonstrates strong scalability. Notably, larger model variants do not overfit the training data but instead achieve consistent performance improvements across diverse benchmarks, as evidenced by the progressive improvements from DSRF-S to DSRF-M and, and from DSRF-M to DSRF-L in Tables 1–3 of the manuscript. The reviewer may refer to the reply to W2/Q2 of Reviewer Yeks for a more comprehensive comparison of DSRF variants across datasets. Due to space limit, we omit here.
>
>
>
> **Q4. Ablation of different mixings in MRM and more**
>
> Thank you for your thoughtful suggestions. We respond to your comments as follows.
>
> - **Ablation of different mixings in MRM**
>
> Following your comments, we have added the ablation of different mixings in MRM, a 3-layers MLP (MLP), two channel-wise 2D convolution layers (CNN), and a self-attention block [e] on O-RVN, I-RVN and RVN-F datasets, as summarized below.
>
> |   | O-RVN | I-RVN | RVN-F |
> | -| - | - | -|
> | w/o MRM | 98.7  | 98.8  | 98.6  |
> | MLP | 98.8  | 98.9  | 98.6  |
> | CNN | 99.0  | 99.2  | 98.8  |
> | Self-Attention | 98.8  | 99.0  | 98.8  |
> | MRM   | 99.1  | 99.4  | 99.0  |
>
> Table D. Ablation of different mixings in MRM.
>
>
>
> The results show that alternative mixings exhibit inferior accuracy compared to the MRM, confirming that our multi-granularity mixing architecture more effectively captures diverse rule interactions. While the improvement is relatively modest compared to other novel architectural components like MVRP and DDCP, the gains persist across dataset while maintaining practical relevance.
>
>
>
> [e] Vaswani, Ashish, et al. "Attention is all you need." *NeurIPS*, 2017.
>
>
>
> - **Additional information on mixings in MRM**
>
> MRM enhances rule composition by performing feature mixing across multiple levels of granularity, including channel-wise, token-wise, row-wise, and sample-wise projections.
>
> - Specifically, the channel-wise mixing is implemented through a linear projection that is applied along the channel dimension, capturing interactions across feature channels and enabling the model to refine semantic dimensions associated with specific attributes.
>
> - For the token-wise mixing, each spatial token is projected using a shared MLP, modelling intra-view spatial relationships and allowing the network to process localized rule patterns within the view.
>
> - For the sample-wise mixing, all tokens within a view are aggregated into a global representation and passed through a linear projection layer, providing global context for each sample and supporting top-down rule verification.
>
> - For the row-wise mixing, features from different matrix rows within a sample are aligned through a linear projection, capturing structural relations such as progression or repetition along the row axis that commonly exist in RPM problems.
>
> These mixing operations capture interactions across different dimensions of the reasoning space, enabling the model to refine and combine partial rule patterns extracted from individual views.
>
>
> We will add these results and discussions in the final manuscript and the supplementary materials accompanying the publication.
>
>
> We trust this rebuttal has comprehensively addressed your concerns. Should any additional questions arise, we would be pleased to provide further clarification.

---

> > ### Comment · Reviewer_Bup5 · 2025-08-09
> >
> > Thank you for the detailed rebuttal and the additional experiments.  My concerns are addressed. I have increased my score.

---

> > > ### Author Response · Authors · 2025-08-09
> > >
> > > We sincerely appreciate the time you have taken to review our rebuttal and provide valuable feedback. Your insightful comments and constructive suggestions have significantly contributed to enhancing the quality of our manuscript.

---

### Author Response · Authors · 2025-08-06
**Gentle Reminder Regarding Our Manuscript Discussion**

We hope this message finds you well. We are writing to kindly follow up on our manuscript <ID: 3366, Title: DSRF: A Dynamic and Scalable Reasoning Framework for Solving RPMs>, which is currently under rebuttal. We fully understand the busy schedules of the reviewers and greatly appreciate the time and effort devoted to providing thoughtful feedback. As the discussion period for our submission is approaching a critical stage, we wish to gently inquire whether there are any further comments after reviewing our rebuttal. Your comments are invaluable for improving our work and ensuring that the discussion can proceed smoothly. Thank you very much for your kind attention and support. We sincerely appreciate your time and consideration, and we look forward to your feedback.

---

> ### Author Response · Authors · 2025-08-08
>
> We hope this message finds you well. As the discussion period for our manuscript (ID: 3366, DSRF: A Dynamic and Scalable Reasoning Framework for Solving RPMs) is drawing to a close, we kindly inquire whether you have any further feedback following your review of our rebuttal. Your insights remain invaluable to us in refining this work, and we deeply appreciate the time and expertise you have contributed to its review process.

---

> > ### Author Response · Authors · 2025-08-09
> >
> > We hope this message finds you well. As the rebuttal period for our manuscript (ID: 3366, Title: DSRF: A Dynamic and Scalable Reasoning Framework for Solving RPMs) approaches its deadline (August 8, 11:59 pm AoE), we would like to extend our sincere gratitude for your insightful comments and the constructive dialogue during this stage. Should you require any further clarification or additional follow-up from our side, please do not hesitate to let us know at your earliest convenience. Your feedback has been invaluable in refining our work, and we deeply appreciate the time and expertise you have dedicated to the review process.

---

### Note · Authors · 2025-08-13

l  **Rebuttal Summary**

We sincerely thank all reviewers for their constructive feedback and for recognizing DSRF’s novel dynamic reasoning (wide, not deep) via MVRP (multi-view reasoning) and DDCP (adaptive task modeling). As acknowledged by **Bup5**, **UTPG** and **Yeks**, the rebuttal comprehensively addresses their key concerns on OOD generalization, design novelty, scalability, data efficiency, and more ablation studies, demonstrating DSRF's dynamic, lightweight, and data-efficient reasoning across benchmarks. After rebuttal, all four reviewers recommended Borderline Accept or better.

The following summarizes the rebuttal for each reviewer.

l  **Bup5 (Borderline Accept or Higher)**

- Our rebuttal addresses: W1: PGM OOD comparisons; W2: Column-wise rule handling; Q1: Sample efficiency on SVRT; Q2: Scoring mechanism; Q3: Multi-view pyramid rationale; Q4: MRM mixing ablation.

- After rebuttal, **Bup5** explicitly stated that “**My concerns are addressed. I have increased my score.**”



l  **UTPG (Borderline Accept)**

- Our rebuttal addresses: W1: Complexity analysis; W2: Data efficiency validation; W3: OOD evaluation criteria; Q1: Held-out testing (PGM/RAVEN); Q2: Generalization to Bongard tasks.

- After rebuttal, **UTPG** explicitly stated that “**These results have addressed most of my earlier concerns**.”

- We also clarify this follow-up inquiry: While OOD gains are modest (due to generalization limits), scaling DSRFs with DDCP's dynamic and hybrid reasoning (row-wise + column-wise) ensures steady progress.



l  **UaAG (Borderline Accept)**

- Our rebuttal addresses: W1: Performance source (novel design vs. simple scaling); W2: DSRF's lightweight scheme; Q1: DSRF-S results (Tables 4-5); Q2: DDCP Gram matrix substitution.

- While we have not yet received an official response from **UaAG**, we are confident that **our rebuttal comprehensively addresses all the reviewer's concerns.**



l  **Yeks (Borderline Accept or Higher)**

- Our rebuttal addresses: W1: Limited novelty despite valid combination; W2/Q2: Scalability analysis; W3/Q3: Unsupported overfitting claim; W4/Q4: Missing LLM comparisons; W5/Q5: Inconsistent paper structure; Q1: MVRP view attribute specificity.

- After rebuttal, **Yeks** explicitly stated that **“The author's response has resolved most of my concerns, and I recommend increasing their score.”**

---

### Decision · Program_Chairs · 2025-09-17

**Decision:**

Accept (poster)

**Comment:**

The paper received all borderline accepts post-rebuttal. The AC, unfamiliar with the subfield, reviewed all materials and recommends acceptance as a poster due to reviewer consensus (Bup5, UTPG, UaAG, Yeks) and score increases from Bup5 and Yeks. Please incorporate promised revisions in the camera-ready version.